# Why Reinforcement Fine-Tuning Preserves Prior Knowledge Better: A Data Perspective

**Zhihao Zhang**[1,2*], **Qiaole Dong**[1*], **Qi Zhang**[1,2,3†],
**Enyu Zhou**[1], **Jun Zhao**[1], **Zhiheng Xi**[1], **Senjie Jin**[1], **Xiaoran Fan**[1], **Yuhao Zhou**[1],
**Mingqi Wu**[1], **Yanwei Fu**[1], **Tao Ji**[1], **Tao Gui**[1,3], **Xuanjing Huang**[1,3†], **Kai Chen**[2†]
[1] Fudan University   [2] Shanghai Artificial Intelligence Laboratory
[3] Shanghai Collaborative Innovation Center of Intelligent Visual Computing
{zhangzhihao19, qldong18, qz}@fudan.edu.cn

## Abstract

Post-training algorithms such as Supervised Fine-Tuning (SFT) and Reinforcement Fine-Tuning (RFT) are widely used to adapt (multimodal) large language models to downstream tasks. While effective at task adaptation, their impact on retaining prior knowledge remains unclear. In this paper, we introduce jigsaw puzzles as a novel task absent from existing pretraining corpora and systematically study the behavior of SFT and RFT on the open-source Qwen2.5-VL series. Our experiments reveal a sharp trade-off: SFT enables rapid task acquisition but leads to catastrophic forgetting, whereas RFT learns more slowly but better maintains prior knowledge. We study this phenomenon through learning dynamics by examining both the magnitude and direction of how training data influence prior knowledge. Our analysis shows that RFT mainly reinforces correct samples naturally aligned with the base model's probability landscape, leading to weaker interference with prior knowledge. Moreover, training on RFT-simulated rollouts, which exert a smaller magnitude of influence and are better aligned in direction to prior knowledge, allows SFT to preserve prior knowledge better while rapidly learning new tasks. We further validate our framework on Qwen2.5 post-training in math and scientific QA, observing consistent forgetting and learning-dynamics trends. These findings suggest that the distribution of post-training data, rather than algorithmic differences alone, plays a central role in forgetting, and highlight RFT as a promising ingredient for stable continual post-training.

## 1 Introduction

In the era of large models, two primary post-training methods, *i.e.*, Supervised Fine-Tuning (SFT) (Wei et al., 2021) and Reinforcement Fine-Tuning (RFT) (DeepSeek-AI et al., 2025; Ouyang et al., 2022), have emerged for enhancing model performance on domain-specific tasks. These methods have been pivotal in enabling (multimodal) large language models to learn specific downstream tasks, follow human instructions, and acquire reasoning capabilities, yielding impressive results. However, current practices primarily focus on performance improvement for specific downstream tasks, while overlooking potential impact of fine-tuning algorithms on model's pre-existing knowledge. This oversight raises concerns about the model's ability to retain and apply prior knowledge.

To this end, this paper investigates how post-training algorithms, specifically SFT and RFT, affect the retention of prior knowledge when large models are trained to learn entirely novel knowledge or tasks that were absent during pretraining. To establish a challenging and genuinely novel task for testing, we introduce jigsaw puzzles as the target task for learning, as in Fig. 1. Through preliminary experiments, we observe that existing state-of-the-art MLLMs, including GPT-4o (Hurst et al., 2024), fail to solve even simple 3x3 jigsaw puzzles, indicating that this task represents a novel problem not covered by current pretraining corpora. Thus, jigsaw puzzles can serve as a fair and meaningful task for evaluating impact of post-training algorithms on prior knowledge.

---

[*] Equal contributions.
[†] Corresponding author.

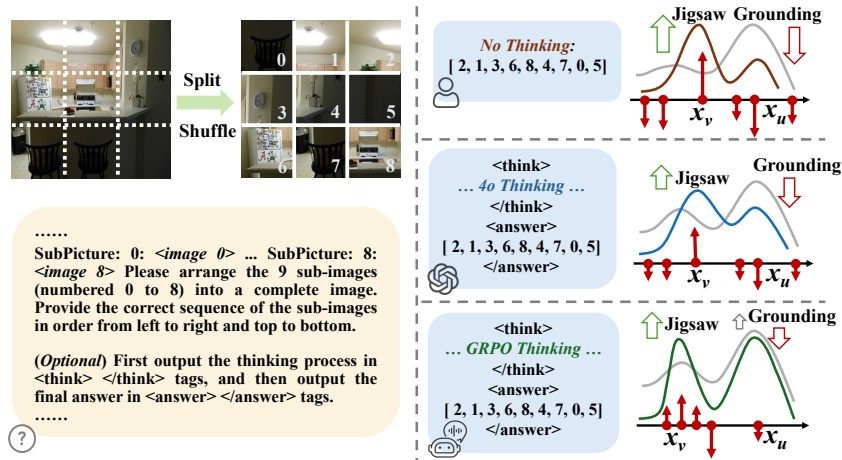

Figure 1: Overview of jigsaw puzzles in the context of MLLMs. We split the original image into 9 patches and randomly shuffle the order of the patches. During SFT, MLLMs are supervised either with Non-Reasoning data directly or GPT-4o-generated reasoning trajectories, while both incur catastrophic forgetting. In contrast, RFT generates reasoning trajectories and answers by itself, reinforces the correct outputs, and avoids severe forgetting.

We conduct systematic fine-tuning experiments using both standard SFT and RFT, *i.e.*, GRPO (Shao et al., 2024), on open-sourced Qwen2.5-VL (Bai et al., 2025) series. Interestingly, we find that SFT can master novel tasks with solely hundreds or thousands of training steps, while RFT requires several tens of thousands of training steps to successfully solve jigsaw puzzles and achieves similar accuracy as SFT. This finding suggests that large-scale RFT can teach model to solve tasks that base model is completely unable to handle. In a sense, this shows that RFT can push the model beyond its original capability boundary. In addition to performance on novel tasks, we observe that SFT incurs severe forgetting of previous knowledge, with substantial performance drops across diverse benchmarks, especially on tasks with similar output formats as jigsaw puzzles. In contrast, RFT, by leveraging reward-driven credit assignments for the simulated rollouts, can master the novel jigsaw puzzles while maintaining decent performance on prior tasks.

The distinct phenomenon between SFT and RFT naturally raises a question: *Why SFT incurs catastrophic forgetting while RFT does not?* While both algorithms increase likelihood of correct responses, RFT adaptively reweights likelihood of rollout with reward, whereas SFT uniformly increases likelihood of static human annotations. Inspired by this, we collect correct rollouts during RFT and use them as supervised data for SFT. Surprisingly, SFT trained on these correct rollouts not only acquires novel knowledge quickly but also preserves prior knowledge better. This suggests that construction of fine-tuning data, rather than training algorithm, is key factor in knowledge forgetting.

Furthermore, we provide a new perspective based on learning dynamics (Ren & Sutherland, 2024), which links the likelihood change of prior knowledge $x_v$ to the gradient induced by an individual training example $x_u$, on understanding this distinct forgetting behavior by analyzing the ***magnitude*** and ***direction*** of how training data influence prior knowledge. We first observe that SFT data without reasoning trajectories usually interfere more with prior knowledge, as verified with a much larger norm of empirical neural tangent kernel (eNTK) between SFT data and prior knowledge. While datasets with reasoning trajectories, such as reasoning trajectories generated by GPT-4o and collected during RFT, usually exhibit a smaller norm of eNTK and less forgetting of prior knowledge, implying that introducing explicit reasoning can help alleviate knowledge forgetting.

As for the direction of interference, we find that data with reasoning trajectories generated by GPT-4o typically belong to high-perplexity regions of the base model. In contrast, data collected during RFT are naturally generated from regions where the base model already assigns a moderate probability. This suggests that pretraining has already shaped certain linguistic regions by accident that are well-aligned with novel tasks, while remaining compatible with prior knowledge. Importantly, according to learning dynamics, the influence of training on one example $x_u$ over the likelihood of another example $x_v$ is symmetric: increasing the likelihood of $x_u$ has the same marginal effect on $x_v$ as increasing $x_v$ has on $x_u$. Therefore, when RFT discovers and reinforces such hidden lin-

guistic regions $x_u$ shaped during pre-training, it degrades less the likelihood of prior knowledge $x_v$. Crucially, such regions are difficult to identify during the stage of dataset construction for SFT, but are accessible through RFT's active exploration within linguistic space. This highlights RFT as an effective algorithm for stable novel knowledge acquisition without suffering from catastrophic forgetting. Further, more experiments on Qwen2.5 post-training in math and scientific QA show consistent forgetting and learning-dynamics results. Formally, our contributions are three-fold:

- We show that RFT can solve unseen tasks while preserving prior competencies. Moreover, SFT trained on RFT-generated rollouts can match RFT's performance while markedly reducing catastrophic forgetting, underscoring the central role of data construction in post-training.

- We propose a learning-dynamics interpretation of forgetting that decomposes how training data influence prior knowledge into its *magnitude* and *direction*, providing a principled view of interference and informing fine-tuning design.

- Building on this interpretation, we conduct extensive experiments demonstrating that RL-sampled corpora strike a favorable magnitude–direction trade-off, offering strong empirical support for the stability of RL algorithms.

## 2 RELATED WORKS

**Jigsaw Puzzles.** Jigsaw puzzles has long been a popular self-supervised task in the computer vision community, aimed at learning visual representations (Noroozi & Favaro, 2016; Carlucci et al., 2019) by spatial reasoning and part-whole understanding. Recently, this task has been repurposed for probing weak spot of MLLMs: Lyu et al. (2025) shows that leading MLLMs perform far behind than human performance. The contemporary work Jigsaw-R1 (Wang et al., 2025) solves jigsaw puzzles with RFT, achieving much better performance. Collectively, these works mainly treat jigsaw puzzles as pretext task for representation learning or test benchmark for MLLMs. However, we employ jigsaw puzzles to investigate how post-training algorithms affect the forgetting behavior of MLLMs.

**Reinforcement Fine-Tuning in MLLMs.** Inspired by the success of RFT in large language models (DeepSeek-AI et al., 2025; Ouyang et al., 2022), recent work has applied RFT to MLLMs. Among them, Meng et al. (2025) finds that RFT can achieve better out-of-distribution generalization performance than SFT. Meanwhile, RFT is also employed for perception-centric tasks (Liu et al., 2025c; Shen et al., 2025; Liu et al., 2025b), still conferring notable gains in generalization and robustness. Concurrently, Jigsaw-R1 (Wang et al., 2025) introduced RFT to the novel task of jigsaw puzzles but achieved limited accuracy. Building on this direction, we extend RFT training to tens of thousands of steps to enable deeper exploration, yielding substantial gains on jigsaw puzzles.

**Catastrophic Forgetting.** Early work (McCloskey & Cohen, 1989; Ratcliff, 1990) showed that even minimal sequential training on disjoint data can cause rapid "catastrophic forgetting" (CF). Existing strategies to mitigate CF fall into three categories: (i) Regularization-based methods (Kirkpatrick et al., 2016; Zenke et al., 2017; Li & Hoiem, 2017) constrain updates to protect old tasks but often limit new learning. (ii) Memory-replay strategies (Shin et al., 2017; Rebuffi et al., 2017; Chaudhry et al., 2019) interleaves past and current data, yet pretraining data of modern open-source MLLMs is usually unavailable for post-training. (iii) Architecture-based techniques (Rusu et al., 2016; Serrà et al., 2018) assign task-specific modules, but their parameter overhead makes them impractical for large MLLMs. Fortunately, with the rise of RFT algorithms such as GRPO (Shao et al., 2024), recent studies (Liu et al., 2025b; Wang et al., 2025; Lai et al., 2025) have shown that RFT can significantly reduce CF in MLLMs, although their analysis and explanations are still limited. Recently, RL's Razor (Shenfeld et al., 2025) argues that online RL mitigates CF because it is implicitly biased toward KL-minimal solutions. Aligning with this discovery, we further analyze why online sampling distribution inherently reduces forgetting from a data-centric perspective with learning dynamics theory. Besides, we would like to clarify that our paper isn't aim to design a better algorithm than classical methods for continuous learning.

## 3 DEFINITION AND BACKGROUND

This section details the format of jigsaw puzzles tailored for MLLMs. For Reinforcement Fine-Tuning (RFT), we propose several rule-based rewards to learn jigsaw puzzles.

### 3.1 Puzzles Generation

**Image Slicing.** Puzzle creation begins with a source image, which is divided into an $m \times n$ regular grid; adjusting $m$ and $n$ directly controls the difficulty of task. If the image height is not divisible by $m$ or the width by $n$, the excess pixels are cropped from the bottom or right edge, respectively, so that the resulting pixels are exact multiples of the grid cell size. The aligned grid is then used to slice the image into $m \times n$ patches, whose order is randomly permuted to produce the puzzle instance.

**Index Assignment and Objective.** To uniquely identify each patch's original position, we assign row-major indices from 0 (top-left) to $m \times n - 1$ (bottom-right). The model receives this permuted sequence of patches as input and must output the indices in canonical top-left to bottom-right order, thereby reconstructing the image. In this study, we adopt a $3 \times 3$ configuration. Empirical results show that state-of-the-art multimodal large language models perform at the chance level on this task.

### 3.2 Rule-based Rewards and RFT

The objective of RFT is driven by a rule-based reward $R$ that comprises three components: the *hit reward* $R_{\text{hit}}$, *accuracy reward* $R_{\text{acc}}$, and *format reward* $R_{\text{fmt}}$.

**Hit Reward.** This term measures partial correctness by computing the fraction of position indices that are predicted accurately:

$$R_{\text{hit}} \;=\; \frac{\# \text{ correct indices}}{m \times n} \;\in\; [0, 1].$$

**Accuracy Reward.** A binary bonus that evaluates whether the entire configuration is correct. The model receives $R_{\text{acc}} = 1$ only when every index is perfectly placed; otherwise $R_{\text{acc}} = 0$.

**Format Reward.** The output must satisfy formatting rules: the reasoning process is wrapped in `<think> ... </think>` tags and the final answer in `<answer> ... </answer>` tags, with each tag appearing exactly once and in the correct order. The final answer must be a non-repeating sequence of digits 0–8 within '[]'. If all requirements are met, $R_{\text{fmt}} = 1$; otherwise, $R_{\text{fmt}} = 0$.

**RFT Algorithm.** We adopt Group Relative Policy Optimization (GRPO) (Shao et al., 2024) as our RFT algorithm. Formally, we maximize the following objective:

$$\mathcal{J}_{\text{GRPO}}(\theta) = \mathbf{E}_{q,\{o_i\}_{i=1}^{G} \sim \pi_{\theta_{\text{old}}}(\cdot|q)} \frac{1}{G} \sum_{i=1}^{G} \frac{1}{|o_i|} \sum_{t=1}^{|o_i|} \left[ \frac{\pi_\theta(o_{i,t}|q, o_{i,<t})}{\pi_{\theta_{\text{old}}}(o_{i,t}|q, o_{i,<t})} A_{i,t} - \beta \mathbf{D}_{\text{KL}}\left(\pi_\theta || \pi_{\text{ref}}\right) \right], \quad (1)$$

where $q$ is the problem, $\mathbf{r} = \{r_1, \cdots, r_G\}$ is reward for model outputs $\{o_1, \cdots, o_G\}$, $A_{i,t} = (r_i - \text{mean}(\mathbf{r}))/\text{std}(\mathbf{r})$ is the advantage for each token. Besides, $\pi_{\theta_{\text{old}}}(\cdot) = \pi_\theta(\cdot)$ in our experiments, so we omit the original *clip* term here for simplicity.

## 4 Experimental Setup

**Dataset Construction.** Jigsaw puzzles are built upon COCO 2014 (Lin et al., 2014) image dataset. For training, we sample around 22k images from the COCO training set to generate jigsaw puzzles. For testing, we sample 100 images from the COCO test set. For the SFT dataset, we provide two data formats, *i.e.*, Non-Reasoning data and Reasoning data: the first directly provides the ground-truth answers without reasoning processes, while the second additionally consists of reasoning trajectories generated by GPT-4o with both the question and the answer as input, dubbed as Rea-4o-Rollout.

**MLLMs.** We employ Qwen2.5-VL-3B (Bai et al., 2025) and Qwen2.5-VL-7B as our MLLMs due to their strong performance on vision-language understanding and support of native resolution input.

**Evaluation.** We not only evaluate the post-trained model on novel tasks, *i.e.*, jigsaw puzzles, but also on 5 representative capability axes of prior knowledge:

- **Grounding.** RefCOCO/+/g (Kazemzadeh et al., 2014; Mao et al., 2016) test referring-expression comprehension, requiring the model to localize objects described by the free-form text.

Table 1: Performance comparison across post-trained models of **Qwen2.5-VL-3B** and **Qwen2.5-VL-7B**. Numbers in parentheses denote the change w.r.t. *each scale's* base model.

| | Qwen2.5-VL-3B | | | | | Qwen2.5-VL-7B | | | | |
|---|---|---|---|---|---|---|---|---|---|---|
| | Base | RFT | SFT-Non-Rea | SFT-Rea-4o-Rollout | SFT-Rea-GRPO-Rollout | Base | RFT | SFT-Non-Rea | SFT-Rea-4o-Rollout | SFT-Rea-GRPO-Rollout |
| *Jigsaw Puzzles (test)* | | | | | | | | | | |
| *Training steps* | – | 27,360 | 200 | 4,100 | 2,670 | – | 27,360 | 400 | 4,100 | 3,000 |
| $3\times3$ puzzles | 0.0 | 66 (↑66) | 53.0 (↑53) | 70.0 (↑70) | 70.0 (↑70) | 0.0 | 75 (↑75) | 80 (↑80) | 78 (↑78) | 81 (↑81) |
| *Grounding* | | | | | | | | | | |
| RefCOCO$_{val}$ | 88.8 | 88.4 (↓0.4) | 6.1 (↓**82.8**) | 74.2 (↓**14.6**) | 84.6 (↓4.2) | 90.0 | 89.4 (↓0.6) | 32.9 (↓**57.2**) | 52.5 (↓**37.5**) | 81.4 (↓8.6) |
| RefCOCO+$_{val}$ | 82.0 | 82.2 (↑0.2) | 4.2 (↓**77.7**) | 68.3 (↓**13.6**) | 77.6 (↓4.4) | 84.7 | 83.6 (↓1.1) | 28.8 (↓**55.9**) | 47.8 (↓**36.9**) | 75.1 (↓9.6) |
| RefCOCOg$_{val}$ | 86.0 | 84.1 (↓1.9) | 5.6 (↓**80.4**) | 71.3 (↓**14.7**) | 80.8 (↓5.2) | 86.4 | 86.3 (↓0.1) | 30.1 (↓**56.4**) | 48.3 (↓**38.1**) | 76.1 (↓10.3) |
| *Document & OCR* | | | | | | | | | | |
| DocVQA$_{test}$ | 92.8 | 91.5 (↓1.3) | 81.6 (↓**11.3**) | 90.3 (↓2.5) | 89.8 (↓3.1) | 94.4 | 94.4 (↑0.0) | 67.1 (↓**27.4**) | 92.1 (↓2.3) | 93.5 (↓0.9) |
| InfoVQA$_{test}$ | 74.3 | 73.1 (↓1.2) | 62.6 (↓**11.7**) | 71.4 (↓2.8) | 70.7 (↓3.6) | 80.1 | 79.1 (↓1.0) | 44.6 (↓**35.5**) | 75.6 (↓4.5) | 77.3 (↓2.8) |
| OCRBench | 79.3 | 77.1 (↓2.1) | 65.9 (↓**13.4**) | 69.4 (↓**9.9**) | 74.9 (↓4.4) | 83.4 | 83.4 (↑0.0) | 51.7 (↓**31.7**) | 80.5 (↓2.9) | 81.4 (↓2.0) |
| *General VQA* | | | | | | | | | | |
| MME$_{sum}$ | 2140 | 2137 (↓3) | 1631 (↓**509**) | 1478 (↓**662**) | 2132 (↓8) | 2333 | 2325 (↓8) | 479 (↓**1854**) | 2084 (↓249) | 2207 (↓126) |
| MMStar | 56.2 | 55.8 (↓0.5) | 49.2 (↓7.0) | 51.7 (↓4.5) | 52.2 (↓4.0) | 62.8 | 64.4 (↑1.7) | 0.0 (↓**62.8**) | 59.1 (↓3.7) | 60.4 (↓2.4) |
| GQA | 60.1 | 59.5 (↓0.6) | 54.7 (↓5.4) | 50.0 (↓**10.1**) | 54.0 (↓6.1) | 60.4 | 60.3 (↓0.1) | 21.7 (↓**38.7**) | 53.5 (↓**6.9**) | 57.0 (↓3.3) |
| *Hallucination* | | | | | | | | | | |
| POPE | 86.9 | 86.5 (↓0.3) | 85.9 (↓1.0) | 69.4 (↓**17.5**) | 85.4 (↓1.4) | 86.2 | 86.0 (↓0.2) | 16.3 (↓**69.9**) | 74.1 (↓**12.1**) | 83.1 (↓3.1) |
| *College-level Problems* | | | | | | | | | | |
| MMMU$_{val}$ | 46.9 | 46.3 (↓0.6) | 43.4 (↓3.5) | 43.0 (↓3.9) | 44.3 (↓2.6) | 51.3 | 50.0 (↓1.3) | 22.4 (↓**28.9**) | 46.1 (↓**5.2**) | 48.8 (↓2.6) |

- **OCR, chart & document understanding.** DocVQA (Mathew et al., 2021), InfoVQA (Mathew et al., 2022), and OCRBench (Liu et al., 2024) probe the ability of MLLMs to read and reason over scanned documents, forms, and scientific plots.

- **General VQA.** MME (Fu et al., 2023), MMStar (Chen et al., 2024), and GQA (Hudson & Manning, 2019) cover visual reasoning, spatial relations, and multimodal commonsense.

- **Hallucination.** POPE (Li et al., 2023) measures tendency of generation not grounded in image.

- **College-level Problems.** MMMU (Yue et al., 2024) is a college-level multimodal benchmark spanning six disciplines for knowledge-grounded visual reasoning.

**Hyper-parameter setup.** All experiments are conducted on $4 \times$ NVIDIA-A800 80GB GPUs. For GRPO tuning on the jigsaw puzzles, we set the number of generations $G{=}4$ per prompt with sampling temperature of 1.0, batch size 4, learning rate $1 \times 10^{-6}$, and KL divergence penalty coefficient $\beta{=}0.04$. For SFT training, we set the batch size to 16 with a learning rate of $1 \times 10^{-5}$ by default.

## 5 RESULTS AND ANALYSIS

### 5.1 CAN RFT MASTER THE NOVEL JIGSAW PUZZLES?

We first test current state-of-the-art multimodal large language models (MLLMs) on our test set of jigsaw puzzles in a zero-shot manner. We find that both GPT-4o and Qwen-2.5-VL-72B obtain an accuracy of 0.0, and their hit rate of correct position indices is close to random chance $1/9$, indicating that jigsaw puzzles are indeed a novel task for these models and are suitable for our research on forgetting during learning of new tasks.

We then examine whether reinforcement fine-tuning can enable the base model to learn entirely new tasks or knowledge from scratch. Specifically, we apply GRPO to Qwen-2.5-VL-3B/7B on the training set of jigsaw puzzles for 10 epochs, encouraging a comprehensive and sufficient exploration of the novel task. After convergence, the final models achieve an accuracy of 66%/75% on the held-out test set as shown in Tab. 1, dramatically outperforming the base model. Qualitative results on test examples, as in Fig. 11 of the Appendix, show that the model learns to generate meaningful reasoning processes before giving the final answers. Although prior work (Yue et al., 2025) suggests that RFT fails to induce fundamentally new reasoning patterns in base models, we show that with sufficiently long-term exploration, RFT can in fact enable the model to solve novel jigsaw puzzles from scratch.

## 5.2 Forgetting of Prior Knowledge: RFT vs. SFT

We also fine-tune Qwen-2.5-VL-3B/7B on the jigsaw puzzle training set using the standard SFT approach with Non-Rea and Rea-4o-Rollout dataset. As shown in Table 1, due to the property of teacher forcing, the model quickly picks up task-specific patterns under SFT, achieving performance comparable to RFT after just one epoch. To assess the impact of SFT and RFT on previously learned knowledge, we further evaluate both models on a set of prior benchmarks in Tab. 1. While SFT achieves high accuracy with much less training time, it leads to significantly more catastrophic forgetting than RFT, even though it is trained for many fewer steps. This forgetting is particularly evident on the Grounding, Document & OCR, and General VQA. Besides, SFT on Non-Rea data incurs much more forgetting than on Rea-4o-Rollout data across several prior benchmarks.

## 5.3 Why does RFT avoid catastrophic forgetting?

We start by analyzing the loss function of RFT and SFT. By carefully comparing the gradients of the RFT and SFT losses (Eq. 11 and Eq. 13, derivation can be found in Appendix C), we find that both losses optimize the model's likelihood. However, the difference lies in the fact that RFT optimizes on the dataset sampled by the model and uses adaptive weights for the likelihood objective, while SFT uniformly improves the model's likelihood on a pre-constructed dataset.

Therefore, we investigate whether the corpus sampled from the model itself enables the base model to learn jigsaw puzzles while retaining its performance on previous tasks with SFT. Specifically, we employ the GRPO-trained model to generate responses on the training split of jigsaw puzzles, and filter the responses based on the correctness of the answer, leaving about 65% of training samples. We then use this filtered corpus to fine-tune the base model under the SFT paradigm. To rule out confounding factors, we adopt exactly the same hyperparameters as in previous SFT experiments.

As shown in Tab. 1, fine-tuning on model-generated data (SFT-Rea-GRPO-Rollout) achieves similar accuracy on jigsaw puzzles, while forgetting much less than SFT-Non-Rea and SFT-Rea-4o-Rollout across most benchmarks. Interestingly, we also find that training on Rea-GRPO-Rollout and Rea-4o-Rollout learns much more slowly than on Non-Rea data, necessitating more training steps to achieve a comparable performance on jigsaw puzzles. This may be because the long reasoning paths dilute the per-token learning signal. Overall, we find that it is not the adaptive weights but the training data that is the key factor why RFT does not suffer from catastrophic forgetting.

## 5.4 Learning Dynamics-based analysis of Data Distribution

Motivated by the observation that the distinct data distributions in the post-training phase lead to different forgetting behaviors, we take a learning dynamics perspective to investigate and explain this phenomenon. Let's consider the SFT loss on different datasets:

$$\min \mathcal{L}(\theta) = -\mathbf{E}_{(q,o,t)\sim\text{Dataset}} \log \pi_\theta(o_t|q, o_{<t}), \qquad (2)$$

where $o_t$ is the ground-truth next token, conditioned on the prompt $q$ and previous completion $o_{<t}$.

Following Ren & Sutherland (2024), we employ learning dynamics to describe "how the change in parameter $\theta$ induced by a step of gradient descent on single training example $x_u \triangleq \{q^u, o^u_{<t}, o^u_t\}$ impacts the probability of another example $x_v \triangleq \{q^v, o^v_{<t}, o^v_t\}$". Here, we treat $x_u$ as a GRPO-rollout sample or man-made SFT sample, and $x_v$ as a sample of prior knowledge. We have

$$\Delta\theta^t(x_u) \triangleq \theta^{t+1} - \theta^t = \eta \cdot \nabla_\theta \log \pi_{\theta^t}(x_u) = \eta \cdot \nabla_\theta \log \pi_{\theta^t}(o^u_t|q^u, o^u_{<t}); \qquad (3)$$

$$\Delta \log \pi^t(x_v)|_{x_u} \triangleq \log \pi_{\theta^{t+1}}(x_v) - \log \pi_{\theta^t}(x_v). \qquad (4)$$

And we want to specify the relationship between $\Delta\theta^t(x_u)$ and $\Delta \log \pi^t(x_v)|_{x_u}$.

**Theorem 5.1.** *Let $\pi_{\theta^t}(x) = \text{Softmax}(\boldsymbol{z}(x))[o_t] \in [0, 1]$, where $\boldsymbol{z}(x) = h_{\theta^t}(q, o_{<t}) \in \mathbb{R}^V$, $V$ is the number of tokens within vocabulary. The one-step learning dynamics has the following format:*

$$\underbrace{\Delta \log \pi^t(x_v)|_{x_u}}_{1\times 1} = \eta \underbrace{\mathcal{A}^t(x_v)}_{1\times V} \underbrace{\mathcal{K}^t(x_v, x_u)}_{V\times V} \underbrace{\mathcal{G}^t(x_u)}_{V\times 1} + \mathcal{O}(\eta^2), \qquad (5)$$

*where $\mathcal{A}^t(x_v) = \nabla_{\boldsymbol{z}} \log \pi_{\theta^t}(x_v)$, $\mathcal{K}^t(x_v, x_u) = (\nabla_\theta \boldsymbol{z}(x_v)|_{\theta^t})(\nabla_\theta \boldsymbol{z}(x_u)|_{\theta^t})^\top$ is the empirical neural tangent kernel (eNTK) of the logit network $\boldsymbol{z}$, and $\mathcal{G}^t(x_u) = \nabla_{\boldsymbol{z}} \log \pi_{\theta^t}(x_u)$.*

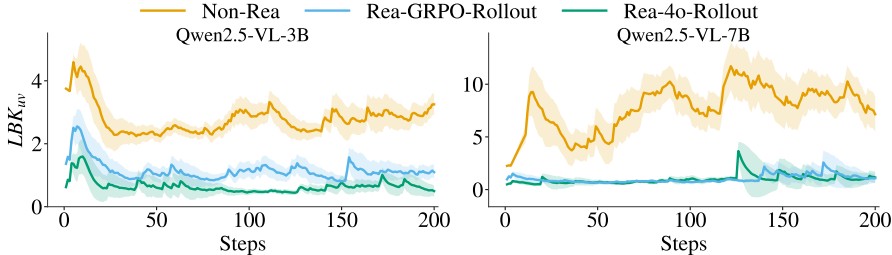

Figure 2: Evolution of LBK$_{uv}$ during the SFT process on three different datasets.

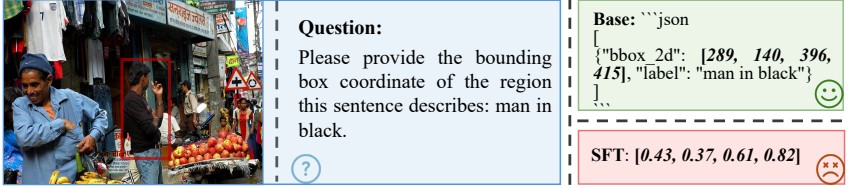

Figure 3: Qualitative result on Grounding before and after training with SFT on Non-Reasoning data. Model finetuned on Non-Reasoning data often switches its output format on Grounding, *i.e.*, from the expected JSON format containing bbox_2d and label to a list of numbers.

Proof of the theorem and more discussion can be found in Appendix D. The theorem shows that the effect of $\Delta\theta^t(x_u)$ on $\Delta\log\pi^t(x_v)|_{x_u}$ is mainly determined by three factors: (1) the model's sensitivity to the old and new knowledge ($\mathcal{A}^t(x_v)$ and $\mathcal{G}^t(x_u)$), and (2) the level of interference between them, captured by $\mathcal{K}^t(x_v, x_u)$. Since the gradients with respect to the logits (i.e., $\mathcal{A}^t(x_v)$ and $\mathcal{G}^t(x_u)$) are typically bounded, this implies that the relative interference is the dominant factor driving forgetting. A larger $||\mathcal{K}^t||_F$ means more interference between $x_u$ and $x_v$. Besides, our analysis in this section also depends on the assumption of "the eNTK matrix $\mathcal{K}^t$ remains roughly stable over training", which is well-validated in Ren & Sutherland (2024) and our following experiments.

So we first measure the interference between the post-training dataset and prior knowledge during the training process of SFT. As it requires huge computation to calculate $||\mathcal{K}^t||_F$ directly, we estimate the Lower Bound of Kernel $||\mathcal{K}^t||_F$ (**LBK**) as follows:

$$||\Delta\log\pi^t(x_v)|_{x_u}||_F \leq \eta||\mathcal{A}^t(x_v)||_F||\mathcal{K}^t(x_v, x_u)||_F||\mathcal{G}^t(x_u)||_F + ||\mathcal{O}(\eta^2)||_F, \tag{6}$$

$$\text{LBK}^t_{uv} \triangleq \frac{||\Delta\log\pi^t(x_v)|_{x_u}||_F^2}{||\mathcal{A}^t(x_v)||_F^2||\mathcal{G}^t(x_u)||_F^2} \lesssim ||\mathcal{K}^t(x_v, x_u)||_F^2. \tag{7}$$

Specifically, we sample responses from base models on Grounding as our prior knowledge $x_v$, which follow a similar answer format to jigsaw puzzles (*i.e.*, with numbers enclosed within '[]') and exhibit the most severe forgetting as in Tab. 1. We then conduct SFT training on three different datasets $x_u$ and record the LBK between prior knowledge and training examples. As shown in Fig. 2, the LBK quickly stabilizes after only a few dozen training steps. Besides, the Non-Reasoning data exhibit much larger LBK compared to the Reasoning data, suggesting stronger interference with prior knowledge. Appendix Fig. 10 further shows that introducing reasoning trajectories improves the model's confidence in answers. These suggest that directly providing answers to new tasks, without linking them to the model's existing perceptual abilities through reasoning trajectories, causes the output distribution to shift abruptly as in Fig. 3, which heavily disrupts prior knowledge and leads to catastrophic forgetting. In contrast, for Reasoning data, the LBK is smaller, meaning that interference with prior knowledge is weaker and forgetting progresses more slowly.

### 5.5 WHAT MAKES THE MODEL-GENERATED REASONING DATA DIFFERENT?

Next, we investigate why reasoning data generated by model itself (Rea-GRPO-Rollout) and by GPT-4o (Rea-4o-Rollout) still result in different forgetting behaviors. To do this, we use the *perplexity* (PPL) of the base model as a measure to compare how well each type of data aligns with the model's distribution. As shown in Fig. 4, Rea-GRPO-Rollout tends to align with the lower-perplexity region of the base model's output distribution, whereas Rea-4o-Rollout typically lies much higher than Rea-GRPO-rollout. This suggests that Rea-GRPO-Rollout is more compatible with the base model's prior knowledge when compared to the Rea-4o-Rollout.

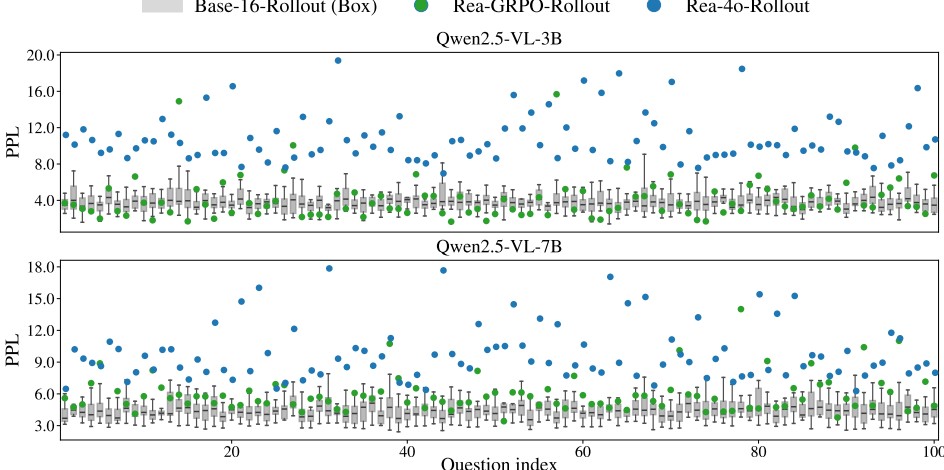

Figure 4: PPL of Rea-GRPO-Rollout and Rea-4o-Rollout under the base model. Base-16-Rollout (Box) denotes PPL range estimated from 16 rollouts generated by base model, serving as a reference.

But then, why does a post-training strategy that focuses on low-perplexity samples alleviate catastrophic forgetting of prior knowledge? Fortunately, we can answer this using the following symmetry property from learning dynamics:

**Theorem 5.2.** *The one-step learning dynamics has the property of symmetry:*

$$\Delta \log \pi^t(x_v)|_{x_u} = \Delta \log \pi^t(x_u)|_{x_v} + \mathcal{O}(\eta^2). \tag{8}$$

According to Theorem 5.2 (Proof in Appendix D), we find that the influence of learning $x_u$ on $x_v$ is nearly the same as the influence of learning $x_v$ on $x_u$. Additionally, since the eNTK matrix stabilizes (Ren & Sutherland, 2024) in the later stages of pretraining, the interactions between $x_u$ and $x_v$ remain consistent across the training step $t$, also observed in Fig. 2. Therefore, models pretrained with prior knowledge show lower perplexity for Rea-GRPO-Rollout, indicating that training with prior knowledge enhances these samples.

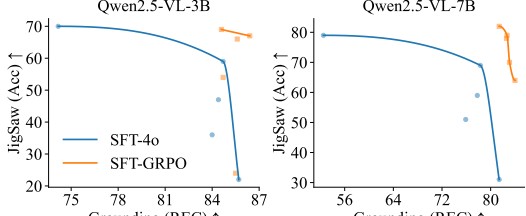

Figure 5: Pareto front curves on the jigsaw and Rec tasks for models fine-tuned on Rea-4o-Rollout and Rea-GRPO-Rollout data.

During post-training, further training on the Rea-GRPO-Rollout samples results in less interference with prior knowledge compared to other higher perplexity samples like Rea-4o-Rollout. As shown in Appendix Fig. 8, when training with Rea-GRPO-Rollout, perplexity of sentences representing prior knowledge continues to decrease on 3B model and remains low on 7B model. In contrast, under Rea-4o-Rollout, perplexity steadily increases. As a result, forgetting effect for Rea-GRPO-Rollout is less pronounced than for Rea-4o-Rollout. Moreover, reinforcement learning algorithms like GRPO, which naturally generate training samples through model rollouts, tend to produce samples with lower perplexity under the base model. This explains why reinforcement learning methods are less prone to catastrophic forgetting.

## 5.6 MORE EXPERIMENTS ON REA-GRPO-ROLLOUT AND REA-4O-ROLLOUT

We further plot the Pareto front curves of accuracy on the Grounding and jigsaw tasks during training on the two datasets. As shown in Fig. 5, the Pareto front from SFT-Rea-GRPO-Rollout is clearly better than that from SFT-Rea-4o-Rollout. Moreover, models trained with Rea-GRPO-Rollout show much smaller performance variance on the Grounding task during the SFT process, indicating that Rea-GRPO-Rollout interferes less with prior knowledge of base models. In contrast, SFT-Rea-4o-Rollout improves jigsaw performance at the cost of degrading Grounding performance. This result further highlights the importance of low-perplexity under the base model as illustrated in Sec. 5.5. Though Rea-4o-Rollout data generally has a smaller LBK on Qwen2.5-VL-3B as in Fig. 2, it still forgets more due to its property of high-perplexity.

Table 2: Performance comparison across post-trained models of **Qwen2.5-3B-Instruct** and **Qwen2.5-7B-Instruct**. Numbers in parentheses denote the change w.r.t. *each scale's* base model.

| | Qwen2.5-3B-Instruct | | | | | Qwen2.5-7B-Instruct | | | | |
|---|---|---|---|---|---|---|---|---|---|---|
| | Base | RFT | SFT-Non-Rea | SFT-Rea-4o-Rollout | SFT-Rea-GRPO-Rollout | Base | RFT | SFT-Non-Rea | SFT-Rea-4o-Rollout | SFT-Rea-GRPO-Rollout |
| *Open-Reasoner-Zero (test) (New Task)* | | | | | | | | | | |
| *Training steps* | – | 2,650 | 1,600 | 2,140 | 2,140 | – | 2,650 | 1,600 | 2,140 | 2,140 |
| ORZ Test | 21.3 | 35.0 (↑13.7) | 23.4 (↑2.1) | 35.4 (↑14.0) | 37.8 (↑16.4) | 32.1 | 49.3 (↑17.2) | 30.3 (↓1.8) | 45.0 (↑12.9) | 53.4 (↑21.3) |
| *Math Reasoning (Old Tasks)* | | | | | | | | | | |
| GSM8k | 84.1 | 83.4 (↓0.7) | 15.1 (↓**69.0**) | 79.9 (↓**9.2**) | 83.0 (↓1.1) | 90.1 | 90.2 (↑0.1) | 21.8 (↓**68.4**) | 85.8 (↓**4.3**) | 90.3 (↑0.2) |
| Math-500 | 42.4 | 55.2 (↑12.8) | 19.4 (↓**23**) | 50.8 (↑8.4) | 54.4 (↑12.0) | 66.6 | 64.8 (↓1.8) | 26.4 (↓**40.2**) | 57.2 (↓**9.4**) | 66.4 (↓0.2) |
| *Instruction Following (Old Task)* | | | | | | | | | | |
| IFEval | 71.6 | 73.4 (↑1.8) | 64.0 (↓**7.6**) | 68.0 (↓3.6) | 72.7 (↑1.1) | 80.6 | 80.5 (↓0.1) | 57.2 (↓**23.4**) | 64.4 (↓**16.2**) | 80.0 (↓0.6) |

Figure 6: Evolution of $LBK_{uv}$ during the SFT process with three different datasets on math dataset.

## 5.7 LLM EXPERIMENTS ON MATH REASONING AND SCIENCEQA

We additionally provide experiments on the LLM Qwen2.5-Instruct (Yang et al., 2024) here, showing its forgetting behavior during post-training on math reasoning, along with the corresponding results. We hope these extra experiments can further strengthen the generality and credibility of our theoretical analysis and conclusions. More detailed experiments setup can be found in Appendix F. As summarized in Tab. 2, the math reasoning experiments exhibit a forgetting pattern highly consistent with our multimodal jigsaw setting: on both 3B and 7B scales, SFT-Non-Rea achieves the largest performance drop on the old math (GSM8K, MATH-500) and instruction-following (IFEval) benchmarks, while reasoning-augmented SFT with external CoT (SFT-Rea-4o-Rollout) forgets less but still substantially more than SFT-Rea-GRPO-Rollout. The latter attains strong gains on the new ORZ task while keeping the performance on old tasks close to the base models, indicating the same hierarchy of forgetting severity, *i.e.*, Non-Rea > Rea-4o > Rea-GRPO.

To probe the underlying mechanism, we compute LBK between post-training samples and prior math knowledge during SFT. As shown in Fig. 6, Non-Rea data consistently display much larger LBK values and Rea-4o also contains occasional high-LBK outliers, providing further evidence for the generality of our learning-dynamics analysis in Sec. 5.4. Moreover, Fig. 7 shows that Rea-4o-Rollout is concentrated in the high-perplexity region of the base models, whereas Rea-GRPO-Rollout lies closer to the low-perplexity region, mirroring our findings on jigsaw puzzles and supporting the low-perplexity training hypothesis in Sec. 5.5 that post-training on model-aligned (low-PPL) reasoning trajectories mitigates catastrophic forgetting. Besides, we also analyze the Pareto front curves (Fig. 13) and Perplexity on prior knowledge (Fig. 14) of different SFT datasets in Appendix F, it is consistent with our theory and prior analysis on jigsaw puzzles.

Beyond math reasoning, we also conduct experiments on a scientific multiple-choice QA benchmark to test the robustness of our conclusions. Concretely, we use the Sci-MCQ4 subset from SciKnow-Eval (Feng et al., 2024); detailed settings and full results are provided in Appendix H. As shown in Tab. 9 and Fig. 16, 17, we again observe the same hierarchy of forgetting severity on prior benchmarks, while SFT-Rea-GRPO-Rollout achieves the best trade-off between performance gains on the new Sci-MCQ4 task and retention on old tasks, further supporting the generality of our analysis.

## 5.8 'COOPERATION' BETWEEN SFT AND RFT

In our previous experiments, we first generate reasoning data after RFT has achieved a high jigsaw accuracy. SFT training on such data can not only achieve high accuracy on the new task, but

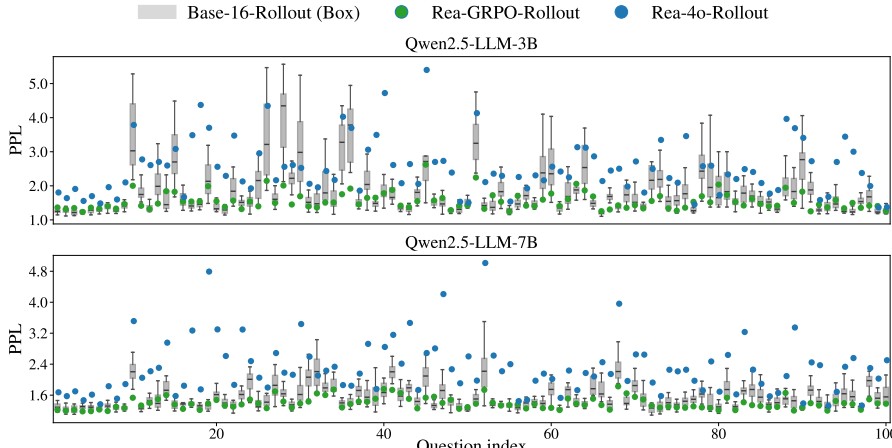

Figure 7: PPL of Rea-GRPO-Rollout and Rea-4o-Rollout math reasoning dataset under the base LLM (Qwen2.5-Instruct). Base-16-Rollout (Box) denotes PPL range estimated from 16 rollouts generated by base model, serving as a reference.

also preserve old knowledge better than Rea-4o-Rollout. This phenomenon suggests that the data distribution is a key factor that determines whether the model forgets during post-training.

To further verify this, we show that if we only want to generate data aligned with the model's own distribution and capable of teaching the new task, it does not require running RFT to very high accuracy. As shown in Tab. 5, we run RFT for only one epoch (5,472 steps), during which the model's jigsaw accuracy stays below 5% (see Fig. 9 (Left) in the appendix). Even so, by collecting the model's rollout CoT and pairing it with the correct answers, we can already construct an effective SFT dataset (Rea-Self-Generated). Fine-tuning the base model on this dataset yields new-task accuracy comparable to RFT and SFT-Rea-GRPO-Rollout, while its performance on old tasks is also similar to them and much better than SFT-Rea-4o-Rollout.

## 6 Discussion and Conclusion

SFT is a widely used post-training method and is often employed as a cold-start phase for RFT (DeepSeek-AI et al., 2025), helping the model acquire basic skills that support subsequent exploration. Besides, SFT also enables the base model to master novel tasks quickly. However, manually curated SFT corpora can lead to the forgetting of prior knowledge. In this work, we show that one can instead construct more stable SFT training data from the model's own reasoning trajectories produced by RFT. Even a short RFT phase is sufficient to generate such self-consistent data (Sec. 5.8), and a subsequent SFT update on this corpus attains new-task performance comparable to RFT while preserving prior knowledge better than SFT-Rea-4o-Rollout. Therefore, developing an efficient and reliable interplay between SFT and RFT that combines their respective advantages remains a promising problem.

This paper provides a systematic investigation into how post-training algorithms affect knowledge retention in multimodal large language models. By introducing jigsaw puzzles as a genuinely novel task, we uncover a clear contrast between SFT and RFT: while SFT enables rapid task acquisition, it suffers from severe forgetting; in contrast, RFT achieves stable learning without significantly degrading prior capabilities. Through empirical studies and theoretical analysis grounded in learning dynamics, we show that this difference arises not from the training algorithm itself, but from the distribution of training data. Specifically, introducing reasonable reasoning trajectories into the SFT process can help alleviate forgetting due to less interference with prior knowledge. Besides, RFT naturally discovers low-perplexity examples that are already partially aligned with the model's output space, making them less disruptive to previous knowledge. Furthermore, using RFT rollouts as supervision enables SFT to forget less, underscoring the importance of fine-tuning data quality. These findings suggest that future post-training efforts should move beyond algorithmic choices and focus more on data selection.

## ACKNOWLEDGMENTS

The authors wish to thank the anonymous reviewers for their helpful comments. This work was partially funded by Henan Province Major Industrial "Challenge-Based Innovation" (No. 251000210300), National Natural Science Foundation of China (No.62476061, 62376061, 62576106, 62521004), Shanghai Rising-Star Program (23QA1400200), and Natural Science Foundation of Shanghai (23ZR1403500). Supported by Shanghai Artificial Intelligence Laboratory.

## ETHICS STATEMENT

From a data distribution perspective, this research employs learning dynamics to explain the advantages of the sampling distribution induced by RL and why RL training tends to yield reduced forgetting. We firmly state that this work is intended for ethical and constructive purposes. Users of this method bear the full responsibility for ensuring it is applied in a safe, fair, and harmless manner. Any misuse of this method is strictly against the intent of the authors.

## REPRODUCIBILITY STATEMENT

We have described our theory analysis and the experiment setup in Sec. 4 and Sec. 5. To support reproducibility, we will open-source our code.

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

## A  LLM USAGE

This article employs large language models solely for polishing the sentence structures to better align with standard English writing conventions.

## B  LIMITATIONS

Due to resource limitations, our experiments are currently restricted to the Qwen-2.5-VL-3B/7B models. In future work, we plan to extend our analysis to larger multimodal models and large language models to assess the generality of our findings. Additionally, this study currently focuses only on the jigsaw puzzle task. Investigating forgetting behaviors across a broader range of multimodal tasks is an important direction we aim to explore next.

## C CONNECTION BETWEEN GRPO AND SFT

This section follows the discussion of DeepSeekMath (Shao et al., 2024) on the unified paradigm of GRPO and SFT closely. And we include the derivation here for the completeness of the paper. We will first derive the gradient of GRPO loss. Specifically, we use the following unbiased estimator as our KL divergence loss:

$$\mathbf{D}_{\mathrm{KL}}\left(\pi_\theta || \pi_{\mathrm{ref}}\right) = \frac{\pi_{\mathrm{ref}}(o_{i,t}|q, o_{i,<t})}{\pi_\theta(o_{i,t}|q, o_{i,<t})} - \log \frac{\pi_{\mathrm{ref}}(o_{i,t}|q, o_{i,<t})}{\pi_\theta(o_{i,t}|q, o_{i,<t})} - 1 \tag{9}$$

By substituting the specific form of the KL divergence into Eq. 1, we get the following function:

$$\mathcal{J}_{\mathrm{GRPO}}(\theta) = \mathbf{E}_{q,\{o_i\}_{i=1}^G \sim \pi_{\theta_{\mathrm{old}}}(\cdot|q)} \frac{1}{G} \sum_{i=1}^G \frac{1}{|o_i|} \sum_{t=1}^{|o_i|} \left[ \frac{\pi_\theta(o_{i,t}|q, o_{i,<t})}{\pi_{\theta_{\mathrm{old}}}(o_{i,t}|q, o_{i,<t})} A_{i,t} \right.$$
$$\left. - \beta \left( \frac{\pi_{\mathrm{ref}}(o_{i,t}|q, o_{i,<t})}{\pi_\theta(o_{i,t}|q, o_{i,<t})} - \log \frac{\pi_{\mathrm{ref}}(o_{i,t}|q, o_{i,<t})}{\pi_\theta(o_{i,t}|q, o_{i,<t})} - 1 \right) \right]. \tag{10}$$

Therefore, the gradient of $\mathcal{J}_{\mathrm{GRPO}}(\theta)$ is:

$$\nabla_\theta \mathcal{J}_{\mathrm{GRPO}}(\theta) = \mathbf{E}_{q,\{o_i\}_{i=1}^G \sim \pi_{\theta_{\mathrm{old}}}(\cdot|q)} \frac{1}{G} \sum_{i=1}^G \frac{1}{|o_i|} \sum_{t=1}^{|o_i|} \left[ \frac{\nabla_\theta \pi_\theta(o_{i,t}|q, o_{i,<t})}{\pi_{\theta_{\mathrm{old}}}(o_{i,t}|q, o_{i,<t})} A_{i,t} \right.$$
$$\left. - \beta \nabla_\theta \left( \frac{\pi_{\mathrm{ref}}(o_{i,t}|q, o_{i,<t})}{\pi_\theta(o_{i,t}|q, o_{i,<t})} - \log \frac{\pi_{\mathrm{ref}}(o_{i,t}|q, o_{i,<t})}{\pi_\theta(o_{i,t}|q, o_{i,<t})} - 1 \right) \right]$$
$$= \mathbf{E}_{q,\{o_i\}_{i=1}^G \sim \pi_{\theta_{\mathrm{old}}}(\cdot|q)} \frac{1}{G} \sum_{i=1}^G \frac{1}{|o_i|} \sum_{t=1}^{|o_i|} \left[ A_{i,t} \nabla_\theta \log \pi_\theta(o_{i,t}|q, o_{i,<t}) \right.$$
$$\left. + \beta \left( 1 - \frac{\pi_\theta(o_{i,t}|q, o_{i,<t})}{\pi_{\mathrm{ref}}(o_{i,t}|q, o_{i,<t})} \right) \frac{\pi_{\mathrm{ref}}(o_{i,t}|q, o_{i,<t})}{\pi_\theta(o_{i,t}|q, o_{i,<t})^2} \nabla_\theta \pi_\theta(o_{i,t}|q, o_{i,<t}) \right] \tag{11}$$
$$= \mathbf{E}_{q,\{o_i\}_{i=1}^G \sim \pi_{\theta_{\mathrm{old}}}(\cdot|q)} \frac{1}{G} \sum_{i=1}^G \frac{1}{|o_i|} \sum_{t=1}^{|o_i|} \left[ A_{i,t} + \right.$$
$$\left. \beta \left( \frac{\pi_{\mathrm{ref}}(o_{i,t}|q, o_{i,<t})}{\pi_\theta(o_{i,t}|q, o_{i,<t})} - 1 \right) \right] \nabla_\theta \log \pi_\theta(o_{i,t}|q, o_{i,<t}),$$

Here, the second equal sign comes from the fact that $\pi_{\theta_{\mathrm{old}}}(\cdot) = \pi_\theta(\cdot)$ in our experiments.

In addition, the SFT objective is to maximize the following format:

$$\mathcal{J}_{\mathrm{SFT}}(\theta) = \mathbf{E}_{q,o \sim \mathrm{Dataset}_{\mathrm{sft}}} \frac{1}{|o|} \sum_{t=1}^{|o|} \log \pi_\theta(o_t|q, o_{<t}). \tag{12}$$

So, the gradient of SFT objective is:

$$\nabla_\theta \mathcal{J}_{\mathrm{SFT}}(\theta) = \mathbf{E}_{q,o \sim \mathrm{Dataset}_{\mathrm{sft}}} \frac{1}{|o|} \sum_{t=1}^{|o|} \nabla_\theta \log \pi_\theta(o_t|q, o_{<t}). \tag{13}$$

Comparing Eq. 11 and Eq. 13, we find that both gradients try to optimize the likelihood of the model. However, they are optimized in different data sources with different gradient coefficients.

## D PROOF OF LEARNING DYNAMICS RELATED THEOREM

**Theorem 5.1.** *Let $\pi_{\theta^t}(x) = \mathrm{Softmax}(z(x))[o_t] \in [0, 1]$, where $z(x) = h_{\theta^t}(q, o_{<t}) \in \mathbb{R}^V$, $V$ is the number of tokens within vocabulary. The one-step learning dynamics has the following format:*

$$\underbrace{\Delta \log \pi^t(x_v)|_{x_u}}_{1 \times 1} = \eta \underbrace{\mathcal{A}^t(x_v)}_{1 \times V} \underbrace{\mathcal{K}^t(x_v, x_u)}_{V \times V} \underbrace{\mathcal{G}^t(x_u)}_{V \times 1} + \mathcal{O}(\eta^2), \tag{5}$$

where $\mathcal{A}^t(x_v) = \nabla_{\boldsymbol{z}} \log \pi_{\theta^t}(x_v)$, $\mathcal{K}^t(x_v, x_u) = (\nabla_\theta \boldsymbol{z}(x_v)|_{\theta^t})(\nabla_\theta \boldsymbol{z}(x_u)|_{\theta^t})^\top$ *is the empirical neural tangent kernel (eNTK) of the logit network* $\boldsymbol{z}$, *and* $\mathcal{G}^t(x_u) = \nabla_{\boldsymbol{z}} \log \pi_{\theta^t}(x_u)$.

*Proof.* We first apply first-order Taylor expansion to approximate $\log \pi_{\theta^{t+1}}(x_v)$ within Eq. 4:

$$\log \pi_{\theta^{t+1}}(x_v) = \log \pi_{\theta^t}(x_v) + \langle \nabla_\theta \log \pi_{\theta^t}(x_v), \, \Delta\theta^t(x_u) \rangle + \mathcal{O}(\|\Delta\theta^t(x_u)\|^2). \quad (14)$$

Then, substituting the gradient descent item (Eq. 3) into the leading term and applying the chain rule of calculus, we get

$$\underbrace{\langle \nabla_\theta \log \pi_{\theta^t}(x_v)}_{1\times d}, \, \underbrace{\Delta\theta^t(x_u) \rangle}_{1\times d} = \big( \underbrace{\nabla_{\boldsymbol{z}} \log \pi_{\theta^t}(x_v)}_{1\times V} \underbrace{\nabla_\theta \boldsymbol{z}(x_v)|_{\theta^t}}_{V\times d} \big) \big( \eta \cdot \underbrace{\nabla_\theta \log \pi_\theta^t(x_u)}_{1\times d} \big)^\top$$

$$= \underbrace{\nabla_{\boldsymbol{z}} \log \pi_{\theta^t}(x_v)}_{1\times V} \underbrace{\nabla_\theta \boldsymbol{z}(x_v)|_{\theta^t}}_{V\times d} \big( \eta \cdot \underbrace{\nabla_{\boldsymbol{z}} \log \pi_{\theta^t}(x_u)}_{1\times V} \underbrace{\nabla_\theta \boldsymbol{z}^t(x_u)|_{\theta^t}}_{V\times d} \big)^\top$$

$$= \eta \underbrace{\nabla_{\boldsymbol{z}} \log \pi_{\theta^t}(x_v)}_{1\times V} \big[ \underbrace{\nabla_\theta \boldsymbol{z}(x_v)|_{\theta^t}}_{V\times d} \underbrace{\big(\nabla_\theta \boldsymbol{z}(x_u)|_{\theta^t}\big)^\top}_{d\times V} \big] \underbrace{\big(\nabla_{\boldsymbol{z}} \log \pi_{\theta^t}(x_u)\big)^\top}_{V\times 1}$$

$$= \eta \mathcal{A}^t(x_v) \mathcal{K}^t(x_v, x_u) \mathcal{G}^t(x_u), \quad (15)$$

where $d$ is the dimension of model parameters $\theta$.

For the remaining second-order term, we should notice that the trick of gradient clip is usually utilized to avoid too large gradients, we have

$$\mathcal{O}(\|\Delta\theta^t(x_u)\|^2) = \mathcal{O}(\eta^2 \|\nabla_\theta \log \pi_{\theta^t}(x_u)\|^2) = \mathcal{O}(\eta^2). \quad (16)$$

Therefore, by reorganizing the terms in Eq. 14, we have

$$\Delta \log \pi^t(x_v)|_{x_u} = \eta \mathcal{A}^t(x_v) \mathcal{K}^t(x_v, x_u) \mathcal{G}^t(x_u) + \mathcal{O}(\eta^2). \qquad \square$$

The second term in this decomposition, $\mathcal{K}^t(x_v, x_u)$, is called the empirical neural tangent kernel and can evolve during training as the network adapts. For sufficiently wide networks initialized properly and trained with small learning rates, $\mathcal{K}^t$ stays nearly fixed throughout training—the limiting kernel in this case is referred to as the neural tangent kernel (Arora et al., 2019; Jacot et al., 2018; Ren & Sutherland, 2024). Additionally, Ren & Sutherland (2024) also validated a relaxed assumption for LLM fine-tuning: the relative influence of learning $x_u$ on other inputs $x_v$ remains roughly stable over training. Besides, the optimization steps during post-training of MLLMs in our paper are very less compared to the steps used in pre-training. So, the relative influence between $x_u$ and $x_v$ during the post-training remains similar to the influence during pre-training is a reasonable hypothesis.

Next, we prove the symmetry theorem of learning dynamics:

**Theorem 5.2.** *The one-step learning dynamics has the property of symmetry:*

$$\Delta \log \pi^t(x_v)|_{x_u} = \Delta \log \pi^t(x_u)|_{x_v} + \mathcal{O}(\eta^2). \quad (8)$$

*Proof.* Following Theorem 5.1, we have

$$\underbrace{\Delta \log \pi^t(x_v)|_{x_u}}_{1\times 1} = \underbrace{\big( \Delta \log \pi^t(x_v)|_{x_u} \big)^\top}_{1\times 1}$$

$$= \big\{ \eta \underbrace{\nabla_{\boldsymbol{z}} \log \pi_{\theta^t}(x_v)}_{1\times V} \big[ \underbrace{\nabla_\theta \boldsymbol{z}(x_v)|_{\theta^t}}_{V\times d} \underbrace{\big(\nabla_\theta \boldsymbol{z}(x_u)|_{\theta^t}\big)^\top}_{d\times V} \big] \underbrace{\big(\nabla_{\boldsymbol{z}} \log \pi_{\theta^t}(x_u)\big)^\top}_{V\times 1} + \mathcal{O}(\eta^2) \big\}^\top$$

$$= \eta \underbrace{\nabla_{\boldsymbol{z}} \log \pi_{\theta^t}(x_u)}_{1\times V} \big[ \underbrace{\nabla_\theta \boldsymbol{z}(x_u)|_{\theta^t}}_{V\times d} \underbrace{\big(\nabla_\theta \boldsymbol{z}(x_v)|_{\theta^t}\big)^\top}_{d\times V} \big] \underbrace{\big(\nabla_{\boldsymbol{z}} \log \pi_{\theta^t}(x_v)\big)^\top}_{V\times 1} + \mathcal{O}(\eta^2)$$

$$= \Delta \log \pi^t(x_u)|_{x_v} - \mathcal{O}(\eta^2) + \mathcal{O}(\eta^2)$$

$$= \Delta \log \pi^t(x_u)|_{x_v} + \mathcal{O}(\eta^2). \qquad \square$$

This theorem points out that the influence of training on $x_u$ over another example $x_v$ is almost similar to the influence of $x_v$ on $x_u$.

---

**Algorithm 1:** Construction Process of the $3 \times 3$ Jigsaw Puzzle Dataset

---

**Input:** Image dataset $\mathcal{D}$ (e.g., COCO-2014); grid size $m = n = 3$
**Output:** Tiles and metadata file
**foreach** $I \in \mathcal{D}$ **do**
    $(H, W) \leftarrow \text{size}(I)$;
    $H' \leftarrow \lceil H/m \rceil \times m$;
    $W' \leftarrow \lceil W/n \rceil \times n$;
    **if** $(H', W') \neq (H, W)$ **then**
        $I' \leftarrow \text{bicubic\_resize}(I, H', W')$;
    **else**
        $I' \leftarrow I$;
    $(h_{\text{tile}}, w_{\text{tile}}) \leftarrow (H'/m, W'/n)$;
    // Row-major slicing into $m \times n$ tiles
    **for** $r \leftarrow 0$ **to** $m - 1$ **do**
        **for** $c \leftarrow 0$ **to** $n - 1$ **do**
            $k \leftarrow r \times n + c$;
            // Crop Tile $k$ from Image
            $T_k \leftarrow \text{crop}(I', r, c, h_{\text{tile}}, w_{\text{tile}})$;
            $\text{save\_image}(T_k)$;
    // Shuffle tiles
    $\pi \leftarrow \text{uniform\_random\_permutation}(\{0, \ldots, m \times n - 1\})$;
    $\text{save\_metadata}(I, H, W, H', W', \pi)$;

---

Table 3: Total training cost (in GPU-hours) for different model sizes and training recipes for the jigsaw puzzles (Qwen2.5-VL-3B/7B) and math reasoning (Qwen2.5-3B/7B).

| Method | Qwen2.5-VL-3B (jigsaw) | Qwen2.5-VL-7B (jigsaw) | Qwen2.5-3B (math) | Qwen2.5-7B (math) |
|---|---|---|---|---|
| RFT | 710 | 2200 | 72 | 96 |
| SFT-Non-Rea | 2.3 | 4 | 0.77 | 1.3 |
| SFT-Rea-4o-Rollout | 6.4 | 11.5 | 1.3 | 2.3 |
| SFT-Rea-GRPO-Rollout | 5.1 | 8.3 | 1.3 | 2.5 |

## E  MORE RESULTS OF JIGSAW PUZZLES

**Jigsaw Dataset Construction Details.** We construct the $3 \times 3$ jigsaw dataset upon MS COCO images with the preprocessing pipeline in Algorithm 1. For each image $I$, we obtain its original size $(H, W)$ and compute the nearest resolution $(H', W')$ such that both $H'$ and $W'$ are divisible by 3, applying bicubic resizing if needed to obtain $I'$. We then partition $I'$ into a $3 \times 3$ grid with tile size $(h_{\text{tile}}, w_{\text{tile}}) = (H'/3, W'/3)$ and assign row-major indices $k \in \{0, \ldots, 8\}$. A **uniform random permutation** $\pi$ of $\{0, \ldots, 8\}$ is used to get the shuffled indices. At training time, the model receives the shuffled tiles and outputs the canonical top-left–to–bottom-right indices.

**Training Cost.** Table 3 summarizes the total GPU-hours (Number of GPU $\times$ Training Hours) for the main experiment configurations of Jigsaw Puzzles and Math Reasoning. The largest configuration (Qwen2.5-VL-7B (jigsaw) RFT) requires about 2200 GPU-hours, while the SFT is two orders of magnitude cheaper.

**Jigsaw Puzzles with Large Learning Rate.** To better illustrate how different training corpora impact forgetting, we increase the learning rate of SFT to $2 \times 10^{-5}$ to amplify the effect of forgetting. As shown in Tab. 4, finetuning on Rea-GRPO-Rollout not only masters the novel task jigsaw puzzles better and faster, but also preserves more prior knowledge than Rea-4o-Rollout. Specifically, finetuning on Rea-4o-Rollout causes severe forgetting after just 300 steps under this larger learning rate, *e.g.*, accuracy on RefCOCO$_{\text{val}}$ drops from 88.8 to 0.16 on Qwen2.5-VL-3B, GQA drops from 60.38 to 42.74 on Qwen2.5-VL-7B. In addition, as training progresses, SFT-Rea-GRPO-Rollout shows slight improvements on some benchmarks, while SFT-Rea-4o-Rollout exhibits a consistent decline on previous benchmarks.

Table 4: Performance of various models on jigsaw puzzles, grounding, document QA, and general VQA benchmarks after SFT with learning rate $2 \times 10^{-5}$.

| Model | Training Steps | Jigsaw-test | Grounding | Document & OCR | General VQA | | | Hallucination |
|---|---|---|---|---|---|---|---|---|
| | | 3×3 puzzles | RefCOCO$_{val}$ | DocVQA$_{test}$ | MME$_{sum}$ | MMStar | GQA | POPE |
| **3B Base** | – | 0 | 88.8 | 92.8 | 2140 | 56.2 | 60.1 | 86.9 |
| **3B Rea-4o-Rollout** | 100 | 7 (↑7) | 0.1 (↓**88.7**) | 90.2 (↓2.6) | 2040 (↓100) | 26.2 (↓**30.0**) | 53.8 (↓6.4) | 82.3 (↓**4.5**) |
| **3B Rea-GRPO-Rollout** | 100 | 23 (↑23) | 81.1 (↓7.7) | 88.3 (↓4.5) | 2014 (↓**126**) | 44.2 (↓12.0) | 47.1 (↓**13.0**) | 86.5 (↓0.4) |
| **3B Rea-4o-Rollout** | 300 | 30 (↑30) | 0.2 (↓**88.6**) | 89.5 (↓3.3) | 1975 (↓**165**) | 25.0 (↓**31.2**) | 50.3 (↓9.8) | 79.0 (↓**7.9**) |
| **3B Rea-GRPO-Rollout** | 300 | 64 (↑64) | 80.5 (↓8.3) | 88.4 (↓4.4) | 2039 (↓**101**) | 42.6 (↓13.6) | 49.7 (↓10.4) | 87.4 (↑0.5) |
| **7B Base** | – | 0 | 90.0 | 94.4 | 2333 | 62.8 | 60.4 | 86.2 |
| **7B Rea-4o-Rollout** | 100 | 29 (↑29) | 67.0 (↓**23.1**) | 93.1 (↓1.3) | 2175 (↓**159**) | 55.9 (↓6.9) | 46.0 (↓**14.4**) | 78.9 (↓**7.3**) |
| **7B Rea-GRPO-Rollout** | 100 | 58 (↑58) | 68.8 (↓**21.2**) | 93.1 (↓1.4) | 2078 (↓**256**) | 55.4 (↓7.4) | 54.6 (↓5.8) | 83.5 (↓2.7) |
| **7B Rea-4o-Rollout** | 300 | 65 (↑65) | 62.0 (↓**28.0**) | 92.9 (↓1.6) | 2062 (↓**272**) | 52.5 (↓**10.3**) | 42.7 (↓**17.6**) | 74.5 (↓**11.7**) |
| **7B Rea-GRPO-Rollout** | 300 | 74 (↑74) | 70.2 (↓19.9) | 93.1 (↓1.3) | 2088 (↓**246**) | 53.5 (↓**9.3**) | 53.9 (↓6.4) | 84.0 (↓2.1) |

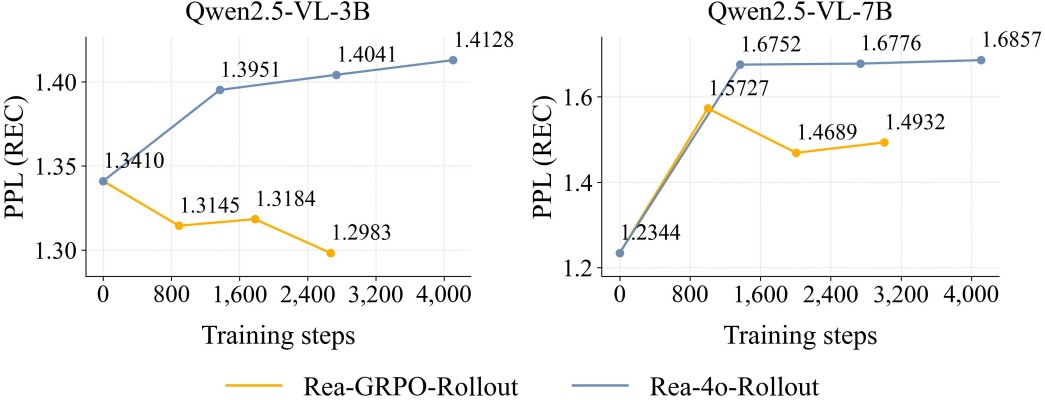

Figure 8: Perplexity versus SFT training steps for Grounding. We collect rollouts from the base model with question from Grounding dataset as our prior knowledge.

**Perplexity on Prior knowledge during SFT.** We further examine how the perplexity of sentences representing prior knowledge changes during SFT. As shown in Fig. 8, with SFT-Rea-GRPO-Rollout, perplexity decreases steadily on the 3B model and stabilizes at a low level on the 7B model (since Rea-GRPO-Rollout data have higher perplexity on 7B than on 3B). In contrast, under SFT-Rea-4o-Rollout, perplexity keeps increasing, highlighting the importance of low-perplexity training data for preserving prior knowledge.

**GRPO with Periodic Restart.** We observe that training with GRPO for too many epochs on a novel task can lead to instability. Periodic restarts are essential for maintaining stable training. As shown in Fig. 9, prolonged training eventually causes the KL divergence loss to explode, consistent with findings from Liu et al. (2025a). To address this, we reset both the optimizer state and the reference policy every 2 epochs. Accordingly, when rule-based constraints or well-shaped reward signals are available, a prolonged exploration stage helps the model generate higher-quality trajectories. This allows it to acquire new skills that are not present in the base model, leading to measurable improvements on the target tasks.

**Answer-Only Perplexity with Reasoning Trajectories.** For each question, we further compute token-level PPL only on the final answers (i.e., the <answer>[...]</answer>part) across three datasets. As shown in Fig. 10, Non-Reasoning data forms a higher, more dispersed band, whereas Rea-GRPO-Rollout and Rea-4o-Rollout data cluster into a lower band across nearly all 100 questions, indicating reasoning trajectories help systematically reducing the uncertainty in decoding answers. Both reasoning dataset exhibit tighter vertical spread than Non-Rea data, suggesting not only lower average PPL but also smaller variance across questions. The observation on PPL across three datasets supports the claim that introducing thinking mitigates conflict from the novel jigsaw objective.

**SFT with self-generated CoTs.** As shown in Tab. 5, using only one epoch of RFT (5,472 steps) to self-generate CoTs and then running SFT (SFT-Rea-Self-Generated) already yields strong jigsaw

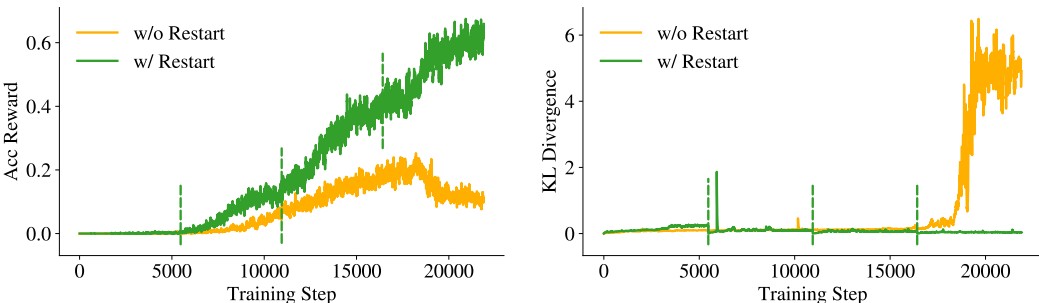

Figure 9: **Left:** Accuracy reward across training steps. **Right:** KL divergence across training steps. The dotted line indicates the point we restart the training of GRPO with newly initialized model.

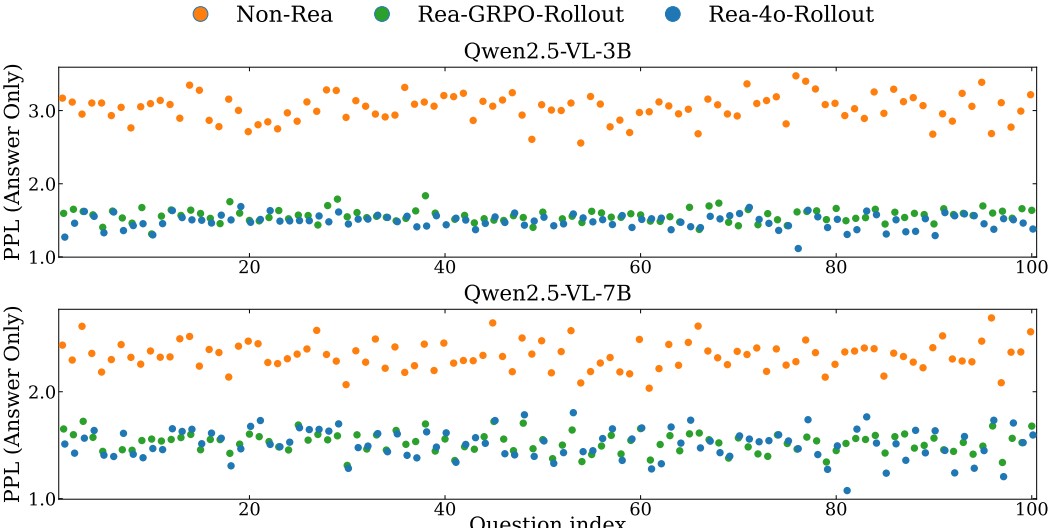

Figure 10: Answer-only perplexity of the numeric sequence (i.e., the <answer>[...]</answer>) for 100 randomly sampled jigsaw items across three datasets. In the Non-Reasoning setting, we insert an empty thinking tag (<think></think>) immediately before the answer.

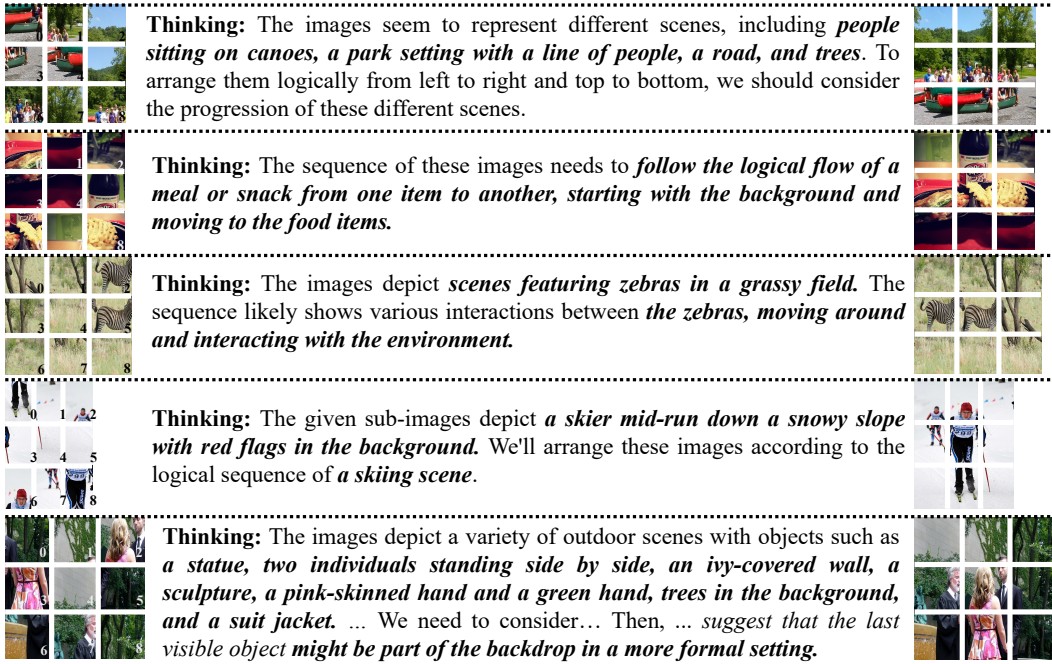

Figure 11: Qualitative Results on jigsaw puzzles (test) after training with RFT.

Table 5: Performance of various post-trained models on jigsaw puzzles, grounding, document QA, and general VQA benchmarks.

| Model | RFT Steps | SFT Steps | Jigsaw-test 3×3 puzzles | Grounding RefCOCO$_{val}$ | Document & OCR DocVQA$_{test}$ | General VQA MME$_{sum}$ | MMStar | GQA | Hallucination POPE |
|---|---|---|---|---|---|---|---|---|---|
| **Qwen2.5-VL-3B** | | | | | | | | | |
| Base | – | – | 0 | 88.8 | 92.8 | 2140 | 56.2 | 60.1 | 86.9 |
| RFT | 27,360 | 0 | 66.0 (↑66) | 88.4 (↓0.4) | 91.5 (↓1.3) | 2137 (↓3) | 55.8 (↓0.5) | 59.5 (↓0.6) | 86.5 (↓0.3) |
| SFT-Rea-4o-Rollout | 0 | 4,100 | 70.0 (↑70) | 74.2 (↓**14.6**) | 90.3 (↓2.5) | 1478 (↓**662**) | 51.7 (↓4.5) | 50.0 (↓**10.1**) | 69.4 (↓**17.5**) |
| SFT-Rea-GRPO-Rollout | 27,360 | 2,670 | 70.0 (↑70) | 84.6 (↓4.2) | 89.8 (↓3.1) | 2132 (↓8) | 52.2 (↓4.0) | 54.0 (↓6.1) | 85.4 (↓1.4) |
| SFT-Rea-Self-Generated | 5,472 | 4,100 | 84.0 (↑84) | 84.5 (↓4.3) | 90.3 (↓2.5) | 2142 (↑2) | 52.4 (↓3.8) | 54.7 (↓5.4) | 88.2 (↑1.3) |
| **Qwen2.5-VL-3B** | | | | | | | | | |
| Base | – | – | 0.0 | 90.0 | 94.4 | 2333 | 62.8 | 60.4 | 86.2 |
| RFT | 27,360 | 0 | 75.0 (↑75) | 89.4 (↓0.6) | 94.4 (↑0.0) | 2325 (↓8) | 64.4 (↓1.7) | 60.3 (↓0.1) | 86.0 (↓0.2) |
| SFT-Rea-4o-Rollout | 0 | 4,100 | 78 (↑78) | 52.5 (↓**37.5**) | 92.1 (↓2.3) | 2084 (↓249) | 59.1 (↓3.7) | 53.5 (↓**6.9**) | 74.1 (↓**12.1**) |
| SFT-Rea-GRPO-Rollout | 27,360 | 3,000 | 81.0 (↑81) | 81.4 (↓8.6) | 93.5 (↓0.9) | 2207 (↓126) | 60.4 (↓2.4) | 57.0 (↓3.3) | 83.1 (↓3.1) |
| SFT-Rea-Self-Generated | 5,472 | 4,100 | 79.0 (↑79) | 86.0 (↓4.0) | 93.8 (↓0.6) | 2256 (↓77) | 60.6 (↓2.2) | 56.7 (↓3.6) | 84.9 (↓1.3) |

performance and largely preserves old-task scores, while avoiding the severe degradation observed in SFT-Rea-4o-Rollout.

**Qualitative jigsaw reconstructions.** Fig. 11 showcases representative test instances solved by the RFT-trained model. The predictions display strong global coherence and semantical consistency, indicating that the model finds globally plausible layouts successfully.

**GPT-4o Prompt and Response.** Fig. 12 shows the prompt we use to elicit GPT-4o "thinking" for the 3 × 3 jigsaw task and a representative response.

## F   MORE DETAILS OF LLM ON MATH REASONING

**Math Reasoning Dataset.** We use the curated math corpus released by Open-Reasoner-Zero (Hu et al., 2025) as our large-scale reasoning-oriented training data, and randomly split it into 90% training and 10% held-out test data. We refer to this held-out split as *ORZ Test*, which serves as a new target task for post-training. Each example is a competition-style math problem paired with a verifiable final answer, without any visual input.

**Prompt for generating thinking of jigsaw tasks.**

```
You are given nine sub-images labeled 0-8 with encodings (
Sub-image 0:  [image]
Sub-image 1:  [image]
Sub-image 2:  [image]
Sub-image 3:  [image]
Sub-image 4:  [image]
Sub-image 5:  [image]
Sub-image 6:  [image]
Sub-image 7:  [image]
Sub-image 8:  [image]) and an optional weak reference
(Reference image (total): [global_image]).  Write one concise
paragraph (no more than 150 words) with planning notes to
help a human consider how these pieces might fit into a 3×3
grid.  Focus only on visual cues such as broad scene features
(e.g., color bands, horizon-like divisions, sky/ground
contrasts), edge continuities (lines, textures, colors
that could extend between tiles), and plausible adjacency
candidates with short justifications using cautious language
(e.g., "likely," "possible," "candidate").  Do not output
the final grid, coordinates, or an ordered list of indices.
Do not reveal or restate any hidden or original arrangement
([GROUND_TRUTH_ORDER]) or provide descriptive details of any
reference image as a whole.  Treat the optional reference
```

```
only as a weak hint and prioritize direct, piece-to-piece edge
analysis.  The goal is to provide helpful assembly guidance
without committing to a definitive solution.  Keep your output
to a single paragraph under 150 words without final ordered
lists.
```

**Example model response.**

To assemble the 3×3 grid, consider visual elements such as the wooden surface, objects on top, and edge alignments. Sub-image 0 features a yellow circular edge that could possibly align with Sub-image 8, given their similar yellow features and adjacent object edges. Sub-image 3, containing onions and chili peppers, seems to share continuity with Sub-image 1 based on onion textures and pepper placements. Sub-image 7, with its mix of objects, could potentially sit near Sub-image 2 due to similarities in scattered textures. The knife-like object in Sub-image 4 might extend toward Sub-image 6, as both show dark, linear edges plausible for alignment. Sub-image 5 has minimal features but might adjoin Sub-image 4 or Sub-image 8, as their backgrounds transition smoothly. Evaluate adjacency by matching subtle overlaps between textures and objects, focusing on edges and angles that continue logically across tiles. Prioritize edge features over object-specific assumptions.

Figure 12: Details of employing GPT-4o to generate reasoning trajectories for jigsaw puzzles.

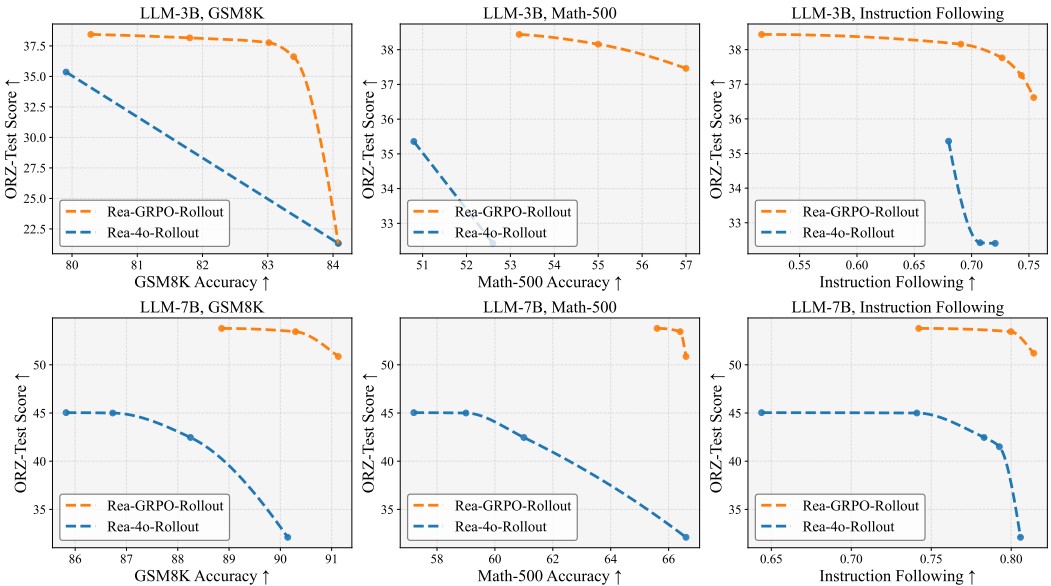

Figure 13: Pareto front curves on the ORZ-test and previous math tasks for models fine-tuned on Rea-4o-Rollout and Rea-GRPO-Rollout data.

**LLMs and Evaluation.** For math experiments, we use Qwen2.5-3B-Instruct (Yang et al., 2024) and Qwen2.5-7B-Instruct as base LLMs. We treat ORZ Test as a new target task and report answer accuracy on it, and use GSM8K (Cobbe et al., 2021), MATH-500 (Hendrycks et al., 2021), and IFEval (Zhou et al., 2023) as prior knowledge to monitor retention of prior math and instruction-following abilities, with their standard accuracy metrics.

**Pareto Frontier of SFT-Rea-GRPO-Rollout and SFT-Rea-4o-Rollout.** We further sweep the learning rate over $\{1 \times 10^{-6}, 5 \times 10^{-6}, 1 \times 10^{-5}, 2 \times 10^{-5}\}$ and, for each setting, plot the Pareto-optimal frontier between performance on the new ORZ task and the performance on old tasks (GSM8K, MATH-500, and IFEval) in Fig. 13. Across all learning rates and both 3B/7B scales, SFT training on Rea-GRPO-Rollout consistently achieves a strictly better Pareto frontier than on Rea-4o-Rollout, yielding either higher ORZ accuracy under a similar level of forgetting, or better retention of prior knowledge at comparable ORZ performance.

**Perplexity on Prior knowledge during SFT.** Fig. 14 also shows that training on Rea-GRPO-Rollout maintains the perplexity of old math corpora much more stably than Rea-4o-Rollout, these results further corroborate our low-perplexity training hypothesis in Sec. 5.5 and demonstrate that the advantages of Rea-GRPO-Rollout are robust under different optimization hyperparameters.

**GRPO Training Recipe.** In our initial experiments, we followed the default settings of the HuggingFace/trl framework, where *num_iterations* is set to 1 (i.e., the GRPO parameter $\mu$). This means that, by default, $\pi_{\theta_{\text{old}}}(\cdot)$ and $\pi_{\theta}(\cdot)$ are identical during training. In Tab. 6, we report RFT results on math reasoning with different GRPO training recipes. The experiments show that using the standard GRPO recipe and using our recipe leads to only minimal differences in model performance on new or old tasks.

**GPT-4o Math Prompt and Response.** Fig. 15 shows the prompt we use to elicit GPT-4o "thinking" for single-step math problems and a representative response.

## G  MIXTURE OF REA AND NON-REA DATA FOR SFT

We have experimented with mixing data of different styles for SFT. Specifically, we combine reasoning and non-reasoning data in equal proportion and apply a unified prompt template: non-reasoning samples produce an empty CoT followed by the answer, while reasoning samples output a CoT first and then the answer. We run this experiment on both the Jigsaw Puzzles and Math Reasoning

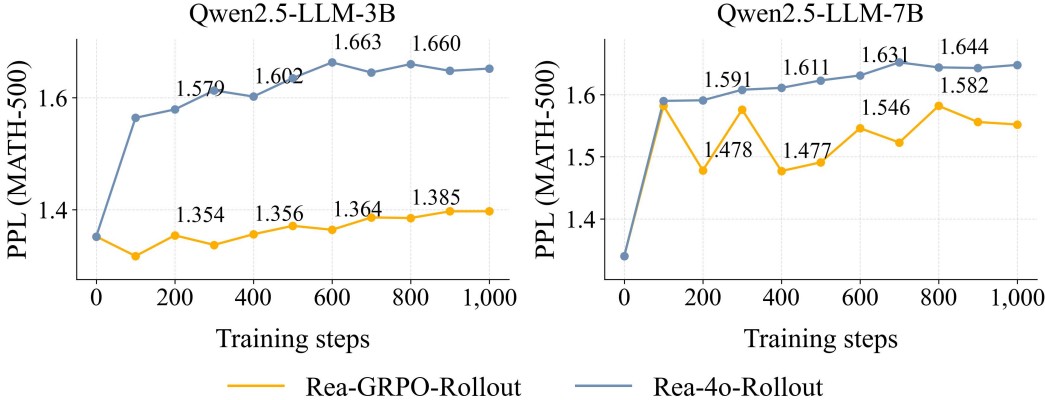

Figure 14: Perplexity versus SFT training steps for MATH-500. We collect rollouts from the base model with question from MATH-500 dataset as our prior knowledge.

Table 6: Performance comparison of different GRPO training recipe.

| | Qwen2.5-3B-Instruct | | | | Qwen2.5-7B-Instruct | | | |
|---|---|---|---|---|---|---|---|---|
| | **Base** | **RFT** ($\mu = 1$) | **RFT** ($\mu = 2$) | **RFT** ($\mu = 4$) | **Base** | **RFT** ($\mu = 1$) | **RFT** ($\mu = 2$) | **RFT** ($\mu = 4$) |
| *Open-Reasoner-Zero (test) (New Task)* | | | | | | | | |
| ORZ Test | 21.32 | 35.00 | 35.70 | 34.10 | 32.11 | 49.29 | 48.18 | 47.12 |
| *Math Reasoning (Old Tasks)* | | | | | | | | |
| GSM8k | 84.08 | 83.40 | 83.02 | 81.27 | 90.14 | 90.22 | 90.29 | 89.69 |
| Math-500 | 42.40 | 55.20 | 55.40 | 52.80 | 66.60 | 64.80 | 65.20 | 64.00 |
| *Instruction Following (Old Task)* | | | | | | | | |
| IFEval | 71.58 | 73.38 | 74.82 | 73.98 | 80.58 | 80.46 | 81.89 | 80.46 |

Table 7: Albation results of Rea and Non-Rea data mixture SFT on jigsaw puzzles.

| Model | Training Steps | Jigsaw-test | Grounding | Document & OCR | General VQA | | | Hallucination |
|---|---|---|---|---|---|---|---|---|
| | | 3×3 puzzles | RefCOCO$_{val}$ | DocVQA$_{test}$ | MME$_{sum}$ | MMStar | GQA | POPE |
| **Qwen2.5-VL-3B** | | | | | | | | |
| **Base** | – | 0 | 88.8 | 92.8 | 2140 | 56.2 | 60.1 | 86.9 |
| **SFT-Non-Rea** | 200 | 53.0 (↑53) | 6.1 (↓**82.8**) | 81.6 (↓**11.3**) | 1631 (↓**509**) | 49.2 (↓7.0) | 54.7 (↓5.4) | 85.9 (↓1.0) |
| **SFT-Rea-4o-Rollout** | 4,100 | 70.0 (↑70) | 74.2 (↓**14.6**) | 90.3 (↓2.5) | 1478 (↓**662**) | 51.7 (↓4.5) | 50.0 (↓**10.1**) | 69.4 (↓**17.5**) |
| **SFT-Rea-GRPO-Rollout** | 2,670 | 70.0 (↑70) | 84.6 (↓4.2) | 89.8 (↓3.1) | 2132 (↓8) | 52.2 (↓4.0) | 54.0 (↓6.1) | 85.4 (↓1.4) |
| **SFT-Mixture** | 1,367 | 70.0 (↑70) | 74.0 (↓**14.8**) | 88.6 (↓4.2) | 1557 (↓**583**) | 46.2 (↓**10.0**) | 46.8 (↓**13.3**) | 70.4 (↓**16.5**) |
| **Qwen2.5-VL-7B** | | | | | | | | |
| **Base** | – | 0.0 | 90.0 | 94.4 | 2333 | 62.8 | 60.4 | 86.2 |
| **SFT-Non-Rea** | 400 | 80.0 (↑80) | 32.9 (↓**57.2**) | 67.1 (↓**27.4**) | 479 (↓**1854**) | 0.0 (↓**62.8**) | 21.7 (↓**38.7**) | 16.3 (↓**69.9**) |
| **SFT-Rea-4o-Rollout** | 4,100 | 78.0 (↑78) | 52.5 (↓**37.5**) | 92.1 (↓2.3) | 2084 (↓**249**) | 59.1 (↓3.7) | 53.5 (↓**6.9**) | 74.1 (↓**12.1**) |
| **SFT-Rea-GRPO-Rollout** | 3,000 | 81.0 (↑81) | 81.4 (↓8.6) | 93.5 (↓0.9) | 2207 (↓**126**) | 60.4 (↓2.4) | 57.0 (↓3.3) | 83.1 (↓3.1) |
| **SFT-Mixture** | 1,367 | 84.0 (↑84) | 26.5 (↓**63.5**) | 93.2 (↓1.2) | 1992 (↓**341**) | 55.7 (↓**7.1**) | 51.4 (↓**9.0**) | 75.4 (↓**10.8**) |

Table 8: Albation results of Rea and Non-Rea data mixture SFT on math reasoning.

| Model | Training Steps | Open-Reasoner-Zero | Math Reasoning | | Instruction Following |
|---|---|---|---|---|---|
| | | ORZ Test | GSM8k | Math-500 | IFEval |
| **Qwen2.5-3B-Instruct** | | | | | |
| **Base** | – | 21.32 | 84.08 | 42.4 | 71.58 |
| **SFT-Non-Rea** | 1,600 | 23.40 (↑2.08) | 15.09 (↓**68.99**) | 19.4 (↓**23**) | 64.03 (↓**7.55**) |
| **SFT-Rea-4o-Rollout** | 2,140 | 35.36 (↑14.04) | 79.91 (↓**9.17**) | 50.8 (↑8.4) | 67.99 (↓3.59) |
| **SFT-Rea-GRPO-Rollout** | 2,140 | 37.76 (↑16.44) | 83.02 (↓1.06) | 54.4 (↑12.0) | 72.66 (↑1.08) |
| **SFT-Mixture** | 2,140 | 32.33 (↑11.01) | 77.86 (↓**6.22**) | 50.6 (↑8.2) | 72.78 (↑1.2) |
| **Qwen2.5-7B-Instruct** | | | | | |
| **Base** | – | 32.11 | 90.14 | 66.6 | 80.58 |
| **SFT-Non-Rea** | 1,600 | 30.32 (↓1.79) | 21.76 (↓**68.38**) | 26.4 (↓**40.2**) | 57.19 (↓**23.39**) |
| **SFT-Rea-4o-Rollout** | 2,140 | 45.04 (↑12.93) | 85.82 (↓**4.32**) | 57.2 (↓**9.4**) | 64.39 (↓**16.19**) |
| **SFT-Rea-GRPO-Rollout** | 2,140 | 53.44 (↑21.33) | 90.30 (↑0.16) | 66.4 (↓0.2) | 79.98 (↓0.6) |
| **SFT-Mixture** | 1,070 | 40.55 (↑8.44) | 72.71 (↓**17.43**) | 54.2 (↓**12.4**) | 73.26 (↓**7.32**) |

datasets. The mixed-data fine-tuning method is denoted as **SFT-Mixture** in the Tab. 7 and Tab. 8. The results show that SFT-Mixture performs between SFT-Non-Rea and SFT-Rea-4o-Rollout, but remains far worse than SFT using model-generated rollout data (SFT-Rea-GRPO-Rollout), even though the latter uses only a single fixed reasoning format. This indicates that simply increasing the stylistic diversity of SFT data does not effectively mitigate catastrophic forgetting. The key is to obtain data that better matches the model's own distribution.

## H  LLM EXPERIMENTS ON SCIENTIFIC MULTIPLE-CHOICE QA

**Scientific MCQ Dataset.** We further investigate catastrophic forgetting on scientific multiple-choice questions using the Sci-MCQ4 subset from SciKnowEval (Feng et al., 2024). From the original corpus, we randomly sample 8,500 examples and split them into 90% training and 10% held-out test data. Each instance is a four-choice science question together with its correct option, covering multi-level scientific knowledge such as physics, chemistry, and biology. We refer to this held-out split as *Sci-MCQ4 Test*, which serves as a target task for post-training.

**LLM and Evaluation.** For scientific QA experiments, we reuse Qwen2.5-3B-Instruct as the base LLM. We treat Sci-MCQ4 Test as the new target task and report answer accuracy on it, and we reuse GSM8K, MATH-500, and IFEval as old tasks with their standard accuracy metrics to monitor retention of prior math and instruction-following abilities.

**Results.** As summarized in Tab. 9, all post-training methods improve performance on the new Sci-MCQ4 task compared to the base model, with SFT-Rea-GRPO-Rollout obtaining the largest gain (+6.6 points). However, the methods exhibit different degrees of forgetting on old tasks: SFT-Non-Rea suffers the most severe degradation on GSM8K (−14.6 points) and also a drop on IFEval, while reasoning-augmented SFT with external CoT (SFT-Rea-4o-Rollout) forgets less but still more than SFT-Rea-GRPO-Rollout. The latter achieves a favorable trade-off, boosting Sci-MCQ4 accuracy while keeping GSM8K, MATH-500 and IFEval close to—or even slightly better than—the base model. This again establishes a consistent hierarchy of forgetting severity, *i.e.*, Non-Rea > Rea-4o > Rea-GRPO, now on a scientific QA benchmark.

---

**Prompt for generating thinking of math tasks.**

```
You will be given a single math problem and the correct final
answer.

Problem:
{PROBLEM}

Correct answer:
{ANSWER}

Write one compact reasoning sketch (no more than 300 words)
that shows a plausible path from the problem statement to
that answer.  Focus on the main ideas:  pattern spotting,
substitutions, identifying recurrences/monotonicity,
modular reductions, or comparing forms until the result
becomes forced.  Use cautious language (``we can see,'' ``it
suggests,'' ``this leads to'') and keep the tone technical.
Do NOT say that the answer was given to you, do NOT restate
the answer at the end, and do NOT list alternative answers.
The goal is to provide hidden working notes that make the
final answer unsurprising.
```

---

**Example model response.**

We consider the nested expression

$$f(x) = \sqrt{1 + x\sqrt{1 + (x+1)\sqrt{1 + (x+2)\sqrt{1 + (x+3)\sqrt{\cdots}}}}}.$$

This suggests defining a recursive form $f(x) = \sqrt{1 + xf(x+1)}$, so that

$$f(x)^2 = 1 + xf(x+1).$$

We can test a simple polynomial candidate $f(x) = x + 1$. Then

$$f(x)^2 = (x+1)^2 = x^2 + 2x + 1, \quad 1 + xf(x+1) = 1 + x(x+2) = x^2 + 2x + 1.$$

The expressions agree, which indicates that $f(x) = x + 1$ is consistent with the recurrence for all positive integers $x$. Since the nested radical is increasing and all terms are positive, this closed form is the stable solution of the recursion. Evaluating at $x = 2008$ gives $f(2008) = 2009$.

---

Figure 15: Details of employing GPT-4o to generate hidden chain-of-thought reasoning trajectories for math problems.

Table 9: Performance comparison across post-trained models of Qwen2.5-3B-Instruct on the scientific multiple-choice dataset **Sci-MCQ4**. Numbers in parentheses denote the change w.r.t. the base model.

| | Steps | Sci-MCQ4 | GSM8K | MATH-500 | IFEval |
|---|---|---|---|---|---|
| Base | – | 65.1 | 84.1 | 42.4 | 71.6 |
| RFT | 960 | 70.8 (↑5.7) | 82.9 (↓1.1) | 45.2 (↑2.8) | 71.8 (↑0.2) |
| SFT-Non-Rea | 240 | 69.3 (↑4.2) | 69.4 (↓**14.6**) | 43.2 (↑0.8) | 70.3 (↓**1.3**) |
| SFT-Rea-4o-Rollout | 214 | 67.0 (↑1.9) | 74.1 (↓**9.9**) | 40.2 (↓2.2) | 73.9 (↑2.3) |
| SFT-Rea-GRPO-Rollout | 214 | 71.7 (↑6.6) | 81.3 (↓2.7) | 40.8 (↓1.6) | 74.5 (↑2.9) |

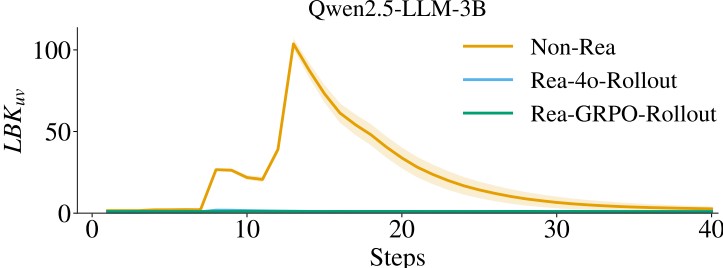

Figure 16: Evolution of $LBK_{uv}$ during the SFT process with three different datasets on the Sci-MCQ4 scientific QA dataset.

**Analysis.** To probe the underlying mechanism, we recompute LBK between post-training samples and prior math knowledge during SFT on Sci-MCQ4. As shown in Fig. 16, Non-Rea data consistently exhibit the largest LBK values, whereas Rea-4o contains occasional high-LBK outliers and Rea-GRPO is concentrated in the low-LBK region, further supporting the learning-dynamics analysis in Sec. 5.4. Moreover, Fig. 17 plots the perplexity of Rea-GRPO-Rollout and Rea-4o-Rollout trajectories under the base LLM. Similar to our findings on math reasoning and multimodal jigsaw puzzles, Rea-4o-Rollout tends to occupy higher-perplexity regions while Rea-GRPO-Rollout stays closer to the low-perplexity band defined by base rollouts, providing additional evidence for the low-perplexity training hypothesis in Sec. 5.5.

**GPT-4o ScienceQA Prompt and Response.** Fig. 18 shows the prompt we use to elicit GPT-4o "thinking" for scientific multiple-choice problems and a representative response.

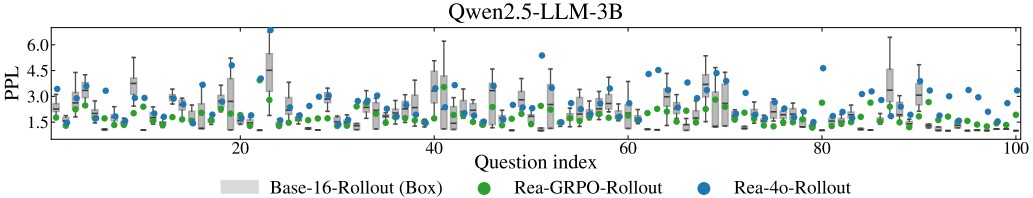

Figure 17: PPL of Rea-GRPO-Rollout and Rea-4o-Rollout trajectories on the Sci-MCQ4 dataset under the base LLM. Base-16-Rollout (Box) denotes the PPL range estimated from 16 rollouts generated by the base model, serving as a reference.

---

**Prompt for generating thinking of Sci-MCQ4 tasks.**

```
You will be given a single math problem and the correct final
answer.

Problem:
{PROBLEM}

Correct answer:
{ANSWER}

Write one compact reasoning sketch (no more than 300 words)
that shows a plausible path from the problem statement to
that answer.  Focus on the main ideas:  pattern spotting,
substitutions, identifying recurrences/monotonicity,
modular reductions, or comparing forms until the result
becomes forced.  Use cautious language (``we can see,'' ``it
suggests,'' ``this leads to'') and keep the tone technical.
Do NOT say that the answer was given to you, do NOT restate
the answer at the end, and do NOT list alternative answers.
The goal is to provide hidden working notes that make the
final answer unsurprising.
```

---

**Example model response.**

The question focuses on the role of NADPH oxidase activation in cardiac dysfunction, specifically its primary function. NADPH oxidase is a key enzyme involved in generating reactive oxygen species (ROS), which are implicated in oxidative stress. Oxidative stress is a well-established contributor to pathological cardiac remodeling and dysfunction. Excessive ROS production can damage cellular components, leading to apoptosis (programmed cell death) and impairing diastolic function by disrupting myocardial relaxation and increasing stiffness.

The activation of NADPH oxidase suggests a pathological mechanism rather than a compensatory or beneficial one, as oxidative stress typically exacerbates cardiac injury rather than promoting hypertrophy or enhancing systolic function. While hypertrophy may occur as a secondary response to stress, the direct link between NADPH oxidase activation and apoptosis aligns with its role in facilitating cellular damage. This indicates that the enzyme's activation is consistent with promoting cardiac apoptosis and diastolic dysfunction, which are hallmarks of oxidative stress-induced cardiac pathology.

Figure 18: Details of employing GPT-4o to generate hidden chain-of-thought reasoning trajectories for Sci-MCQ4 problems.

