# OpenReview forum: "Why Reinforcement Fine-Tuning Enables MLLMs Preserve Prior Knowledge Better: A Data Perspective"
_ICLR.cc/2026/Conference — ICLR 2026 Poster_

### Official Review · Reviewer_DcMQ · 2025-10-26

**Soundness:** 2
**Presentation:** 3
**Contribution:** 2
**Rating:** 4
**Confidence:** 4

**Summary:**

This paper investigates the differences between Supervised Fine-Tuning (SFT) and Reinforcement Fine-Tuning (RFT) regarding catastrophic forgetting. The authors find that RFT learns new knowledge more slowly but significantly reduces catastrophic forgetting, while SFT learns faster but forgets old knowledge more quickly. To study this, the paper introduces a jigsaw puzzle task, which is novel to existing models. The authors analyze how RFT and SFT perform on this task and conclude as above. They also find that using reasoning trajectories generated by the fine-tuned model itself (e.g., Qwen2.5-VL) for SFT—rather than those generated by other models like GPT-4o—greatly mitigates catastrophic forgetting, while also retaining higher training efficiency than RFT. This leads to a balanced approach between learning speed and knowledge retention.

**Strengths:**

- The paper provides a comprehensive comparison between RFT and SFT in both learning efficiency and resistance to catastrophic forgetting, yielding valuable insights.
- It proposes a method that uses model-generated reasoning trajectories for SFT, achieving both faster learning and stronger retention of prior knowledge.

**Weaknesses:**

- All conclusions are drawn from experiments on the jigsaw puzzle task using only the Qwen2.5-VL family. This setting is rather narrow and makes it difficult to claim general applicability. Therefore, the current experimental scope may not fully guarantee that the conclusions hold across other models or tasks.
- Although Rea-GRPO-Rollout converges faster than standard RFT during training, it requires an additional stage of RFT training and reasoning trajectory generation beforehand. Hence, the overall time cost (RFT training + data preparation + SFT) is likely higher, which may reduce its practical appeal.
- The poor performance of SFT-Non-Rea might partly stem from the overly uniform format of its answers, rather than SFT itself. If the training data included more diverse question and answer styles, the extent of catastrophic forgetting might be reduced.

**Questions:**

- Have the authors experimented with other models or tasks beyond jigsaw puzzles? Even partial results on different setups could strengthen the generality and credibility of the conclusions.
- Have the authors examined whether using SFT data with more stylistic diversity (e.g., mixing reasoning-based, multiple-choice, direct-answer, or natural-language descriptions of the answer) could also alleviate catastrophic forgetting?

---

> ### Author Response · Authors · 2025-11-20
> **Comment of Reviews by Reviewer DcMQ [1/3]**
>
> ### **Weaknesses** ###
>
> **Weaknesses 1:**
>
> Thanks. Please refer to our **General Comment** for More Experiments on Other Tasks and Models.
>
> **Weaknesses 2:**
>
> Thanks. First, we want to clarify that our goal is only to use SFT on the rollouts collected after GRPO training to verify that the *data distribution* is the key factor determining whether the model forgets during post-training. We are not claiming that an “RFT → SFT” pipeline is more efficient than RFT.
>
> Second, our further experiments show that generating data aligned with the model’s own distribution, which is also capable of teaching the model the new task, does not require running RFT to very high accuracy. As shown in the table below, we run RFT for only one epoch (5,472 steps), during which the model’s jigsaw accuracy stays below 5% (see **Fig. 9** (left) in the appendix). Even so, by collecting the model’s rollout CoT and pairing it with the correct answers, we can already construct an effective SFT dataset (Rea-Self-generated). Fine-tuning the base model on this dataset yields new-task accuracy comparable to RFT and SFT-Rea-GRPO-Rollout, while its performance on old tasks is also similar to them, much better than SFT-Rea-4o-Rollout. This demonstrates that the overall training pipeline has strong potential and does not require a very high computational cost indeed.
>
> **Table: Performance of various post-trained models on jigsaw puzzles, grounding, document QA, and general VQA benchmarks.**
> | Backbone       | Model                    | RFT Steps | SFT Steps | 3x3 puzzles | RefCOCO_val | DocVQA_test | MME_sum | MMStar | GQA   | POPE  |
> |----------------|--------------------------|-----------|-----------|-------------|-------------|-------------|---------|--------|-------|-------|
> | Qwen2.5-VL-3B  | Base                     | --        | --        | **0**       | **88.8**    | **92.8**    | **2140**| **56.2** | **60.1** | **86.9** |
> | Qwen2.5-VL-3B  | RFT                      | 27,360    | 0         | 66.0        | 88.4        | 91.5        | 2137    | 55.8   | 59.5  | 86.5  |
> | Qwen2.5-VL-3B  | SFT-Rea-4o-Rollout       | 0         | 4,100     | 70.0        | 74.2        | 90.3        | 1478    | 51.7   | 50.0  | 69.4  |
> | Qwen2.5-VL-3B  | SFT-Rea-GRPO-Rollout     | 27,360    | 2,670     | 70.0        | 84.6        | 89.8        | 2132    | 52.2   | 54.0  | 85.4  |
> | Qwen2.5-VL-3B  | SFT-Rea-Self-Generated   | 5,472     | 4,100     | 84.0        | 84.5        | 90.3        | 2142    | 52.4   | 54.7  | 88.2  |
> | Qwen2.5-VL-7B  | Base                     | --        | --        | **0.0**     | **90.0**    | **94.4**    | **2333**| **62.8** | **60.4** | **86.2** |
> | Qwen2.5-VL-7B  | RFT                      | 27,360    | 0         | 75.0        | 89.4        | 94.4        | 2325    | 64.4   | 60.3  | 86.0  |
> | Qwen2.5-VL-7B  | SFT-Rea-4o-Rollout       | 0         | 4,100     | 78.0        | 52.5        | 92.1        | 2084    | 59.1   | 53.5  | 74.1  |
> | Qwen2.5-VL-7B  | SFT-Rea-GRPO-Rollout     | 27,360    | 3,000     | 81.0        | 81.4        | 93.5        | 2207    | 60.4   | 57.0  | 83.1  |
> | Qwen2.5-VL-7B  | SFT-Rea-Self-Generated   | 5,472     | 4,100     | 79.0        | 86.0        | 93.8        | 2256    | 60.6   | 56.7  | 84.9  |

---

> ### Author Response · Authors · 2025-11-20
> **Comment of Reviews by Reviewer DcMQ [2/3]**
>
> **Weaknesses 3:**
>
> Thanks. We conducted ablation experiments using *SciKnowEval*[1], a scientific question-answering dataset covering multiple disciplines and diverse question types, including content generation, multiple-choice questions, relation extraction questions, and true/false questions. From each type, we sampled up to 2,000 questions, forming a mixed-style dataset of around 8,000 examples, referred to as Non-Rea (**Diverse Style**). For the single-style setup, we used only 8,000 multiple-choice questions as the training set, dubbed as Non-Rea (**Single Style**).
>
> Additionally, we held out a separate set of multiple-choice questions, disjoint from the training data, to serve as the test set for evaluating new knowledge acquisition in both experimental groups. For assessing preservation of old knowledge, we used established benchmarks in math reasoning and instruction following.
>
> We tested multiple learning rates across all datasets. As shown in the table below, incorporating diverse question styles does not effectively mitigate catastrophic forgetting caused by Non-Rea data; in fact, Non-Rea (Diverse Style) sometimes even performs worse than Non-Rea (Single Style).
>
> Finally, we performed RFT using only multiple-choice data. The results show that both SFT-Non-Rea (Diverse Style) and SFT-Non-Rea (Single Style) exhibit significantly more forgetting than the RFT model. This further supports the effectiveness of RFT in alleviating knowledge forgetting during post-training.
>
> **Table: Performance of 3B models with SFT training on Diverse /Single Style tasks.**
> | Category    | Model                   | SQA_test_mcq4 (IND) | GSM8K (OOD) | MATH-500 (OOD) | IFEval (OOD) |
> |-------------|-------------------------|---------------------:|------------:|---------------:|-------------:|
> | **Base**    | **3B-Base**             | **65.094**           | **84.08**   | **42.4**       | **0.7158**   |
> |**RFT-Single**| **3B-RFT-Single-lr1e-6**| **70.754**           | **82.941**  | **45.2**       | **0.7182**   |
> | SFT-Diverse | 3B-SFT-Diverse-lr1e-6   | 67.452               | 57.012      | 32.0           | 0.6775       |
> | SFT-Diverse | 3B-SFT-Diverse-lr5e-6   | 71.226               | 59.287      | 41.6           | 0.5264       |
> | SFT-Diverse | 3B-SFT-Diverse-lr1e-5   | 72.169               | 42.077      | 28.0           | 0.3585       |
> | SFT-Single  | 3B-SFT-Single-lr1e-6    | 69.339               | 69.446      | 43.2           | 0.7026       |
> | SFT-Single  | 3B-SFT-Single-lr5e-6    | 75.471               | 15.163      | 13.2           | 0.6127       |
> | SFT-Single  | 3B-SFT-Single-lr1e-5    | 77.830               | 13.798      | 10.8           | 0.4305       |
>
> [1] *Kehua Feng, Keyan Ding, Weijie Wang, Xiang Zhuang, Zeyuan Wang, Ming Qin, Yu Zhao, Jianhua Yao, Qiang Zhang, and Huajun Chen. Sciknoweval: Evaluating multi-level scientific knowledge of large language models. arXiv preprint arXiv:2406.09098, 2024.*

---

> ### Author Response · Authors · 2025-11-20
> **Comment of Reviews by Reviewer DcMQ [3/3]**
>
> ### **Questions** ###
>
> **Question 1:**
>
> Thanks. Please refer to our **General Comment** for More Experiments on Other Tasks and Models.
>
> **Question 2:**
>
> Thanks, we have experimented with mixing data of different styles for SFT. Specifically, we combine reasoning and non-reasoning data in equal proportion and apply a unified prompt template: non-reasoning samples produce an empty CoT followed by the answer, while reasoning samples output a CoT first and then the answer. We run this experiment on both the Jigsaw Puzzles and Math Reasoning datasets. The mixed-data fine-tuning method is denoted as **SFT-Mixture** in the tables below.
>
> The results show that SFT-Mixture performs between SFT-Non-Rea and SFT-Rea-4o-Rollout, but remains far worse than SFT using model-generated rollout data (SFT-Rea-GRPO-Rollout), even though the latter uses only a single fixed reasoning format. This indicates that simply increasing the stylistic diversity of SFT data does not effectively mitigate catastrophic forgetting. The key is to obtain data that better matches the model’s own distribution.
>
> **Table: Albation results of Rea and Non-Rea data mixture SFT on jigsaw puzzles.**
> | Backbone       | Model                 | Training Steps | 3x3 puzzles | RefCOCO_val | DocVQA_test | MME_sum | MMStar | GQA   | POPE  |
> |----------------|-----------------------|----------------|-------------|-------------|-------------|---------|--------|-------|-------|
> | Qwen2.5-VL-3B  | Base                  | --             | **0**       | **88.8**    | **92.8**    | **2140**| **56.2** | **60.1** | **86.9** |
> | Qwen2.5-VL-3B  | SFT-Non-Rea           | 200            | 53.0        | 6.1         | 81.6        | 1631    | 49.2   | 54.7  | 85.9  |
> | Qwen2.5-VL-3B  | SFT-Rea-4o-Rollout    | 4,100          | 70.0        | 74.2        | 90.3        | 1478    | 51.7   | 50.0  | 69.4  |
> | Qwen2.5-VL-3B  | SFT-Rea-GRPO-Rollout  | 2,670          | 70.0        | 84.6        | 89.8        | 2132    | 52.2   | 54.0  | 85.4  |
> | Qwen2.5-VL-3B  | SFT-Mixture           | 1,367          | 70.0        | 74.0        | 88.6        | 1557    | 46.2   | 46.8  | 70.4  |
> | Qwen2.5-VL-7B  | Base                  | --             | **0.0**     | **90.0**    | **94.4**    | **2333**| **62.8** | **60.4** | **86.2** |
> | Qwen2.5-VL-7B  | SFT-Non-Rea           | 400            | 80.0        | 32.9        | 67.1        | 479     | 0.0    | 21.7  | 16.3  |
> | Qwen2.5-VL-7B  | SFT-Rea-4o-Rollout    | 4,100          | 78.0        | 52.5        | 92.1        | 2084    | 59.1   | 53.5  | 74.1  |
> | Qwen2.5-VL-7B  | SFT-Rea-GRPO-Rollout  | 3,000          | 81.0        | 81.4        | 93.5        | 2207    | 60.4   | 57.0  | 83.1  |
> | Qwen2.5-VL-7B  | SFT-Mixture           | 1,367          | 84.0        | 26.5        | 93.2        | 1992    | 55.7   | 51.4  | 75.4  |
>
> **Table: Albation results of Rea and Non-Rea data mixture SFT on math reasoning.**
> | Backbone              | Model                  | Training Steps | ORZ Test | GSM8k   | Math-500 | IFEval  |
> |-----------------------|------------------------|----------------|----------|---------|----------|---------|
> | Qwen2.5-3B-Instruct   | Base                   | --             | **21.32** | **84.08** | **42.4**  | **71.58** |
> | Qwen2.5-3B-Instruct   | SFT-Non-Rea            | 1,600          | 23.40    | 15.09   | 19.4     | 64.03   |
> | Qwen2.5-3B-Instruct   | SFT-Rea-4o-Rollout     | 2,140          | 35.36    | 79.91   | 50.8     | 67.99   |
> | Qwen2.5-3B-Instruct   | SFT-Rea-GRPO-Rollout   | 2,140          | 37.76    | 83.02   | 54.4     | 72.66   |
> | Qwen2.5-3B-Instruct   | SFT-Mixture            | 2,140          | 32.33    | 77.86   | 50.6     | 72.78   |
> | Qwen2.5-7B-Instruct   | Base                   | --             | **32.11** | **90.14** | **66.6**  | **80.58** |
> | Qwen2.5-7B-Instruct   | SFT-Non-Rea            | 1,600          | 30.32    | 21.76   | 26.4     | 57.19   |
> | Qwen2.5-7B-Instruct   | SFT-Rea-4o-Rollout     | 2,140          | 45.04    | 85.82   | 57.2     | 64.39   |
> | Qwen2.5-7B-Instruct   | SFT-Rea-GRPO-Rollout   | 2,140          | 53.44    | 90.30   | 66.4     | 79.98   |
> | Qwen2.5-7B-Instruct   | SFT-Mixture            | 1,070          | 40.55    | 72.71   | 54.2     | 73.26   |

---

> ### Author Response · Authors · 2025-11-27
> **Invitation to discussion**
>
> Dear Reviewer DcMQ,
>
> Thank you again for your thoughtful review. To address your concerns, we:
> - Added **new experiments on Qwen2.5-3B/7B-Instruct (math: ORZ, GSM8K, MATH-500, IFEval)**, which demonstrate that our findings go beyond multimodal jigsaw.
> - Proposed a more efficient **RFT → SFT-Rea-Self-Generated** pipeline to obtain model-aligned rollouts without very expensive RFT.
> - Conducted **SFT-Non-Rea (diverse vs single style)** and **SFT-Mixture** experiments, confirming that *data distribution/model alignment*, rather than stylistic diversity alone, is key to mitigating forgetting.
> - Updated the PDF, where **all new or revised content is highlighted in purple** for easy checking.
>
> We hope these responses have addressed your concerns and would appreciate it if you could consider raising your final rating. We look forward to your further feedback!
>
> Best regards,
> The Authors

---

### Official Review · Reviewer_sRUL · 2025-10-26

**Soundness:** 2
**Presentation:** 3
**Contribution:** 3
**Rating:** 4
**Confidence:** 3

**Summary:**

This paper systematically investigate the impact of supervised fine-tuning (SFT) and reinforcement fine-tuning (RFT) on prior knowledge retention in multimodal large language models (MLLMS) by introducing a jigsaw puzzle task as a novel learning scenario.    This paper present an interesting and important finding: RFT outperforms SFT in avoiding catastrophic forgetting, and further provide a theoretical explanation in terms of data distribution and learning dynamics.

**Strengths:**

1、This paper explicitly address a problem that is crucial at the intersection of Continual Learning (CL) and Reinforcement Learning (RL) : how to avoid catastrophic forgetting when adapting to a new task.
2、Based on the theoretical analysis of Learning Dynamics and eNTK, this paper explain the impact of different data distributions on forgetting from two dimensions of magnitude and direction, which provides a new perspective for understanding the advantages of RFT.

**Weaknesses:**

1. Although the paper studies "catastrophic forgetting", the discussion of existing continuous learning (CL) methods is relatively superficial, and the classical CL methods (e.g., EWC, LwF, ER, etc.) are not fully compared.
2. While the authors note that resource constraints prevented experimentation with more multimodal models and large language models, experimenting on more methods is key to improving the contribution of this work.
3. The learning dynamic analysis formulation in Section 5.4 is dense and not friendly enough for readers with a non-theoretical background.
4. The calculation method of A_(i,t) is not defined in formula (1).

**Questions:**

1. RFT requires ~ 27k training steps while SFT only takes a few hundred, which may lead to an unfair comparison (different amount of training)?
2. The paper assumes that eNTK remains stable during training, but it is questionable whether this holds in MLLM fine-tuning?
3. Is the generation prompt of the "Rea-4o-Rollout" data in Section 4 consistent with the inference process generated by RFT?  If not, does it affect the fairness of the comparison?

---

> ### Author Response · Authors · 2025-11-20
> **Comment of Reviews by Reviewer sRUL**
>
> ### **Weaknesses** ###
> **Weaknesses 1:**
>
> Thanks, we would like to clarify that this paper focuses on understanding why RFT preserves prior knowledge better than SFT in the post-training era of large models, rather than proposing a new or improved continual learning algorithm. Therefore, we donot compare RFT or SFT to classical CL methods in our paper. Our analysis centers on the algorithmic differences between RFT and SFT, and we want to develop an interpretability framework from a data-distribution perspective for understanding the forgetting behaviour of post-training algorithms.
>
> Besides, we have added and cited classic continual learning methods (e.g., *EWC, LwF, ER, etc.*) in the revised version of the paper and have explicitly clarified this positioning.
>
> **Weaknesses 2:**
>
> Thanks, Please refer to our **General Comment** for More Experiments on Other Tasks and Models.
>
> **Weaknesses 3:**
>
> Thanks, we have revised **Sec. 5.4** and added more explanatory text to clarify the meaning of **Theorem 5.1**. Our goal is to make this section easier to understand for readers without a theoretical background. The added text is as follows:
>
> "*The theorem shows that the effect of $\Delta\theta^t(\textcolor{red}{x_u})$ on $\Delta\log\pi^t(\textcolor{blue}{x_v})|_{\textcolor{red}{x_u}}$ is mainly determined by three factors: (1) the model’s sensitivity to the old and new knowledge ($\mathcal{A}^t(\textcolor{blue}{x_v})$ and $\mathcal{G}^t(\textcolor{red}{x_u})$), and (2) the level of interference between them, captured by $\mathcal{K}^t(\textcolor{blue}{x_v},\textcolor{red}{x_u})$. Since the gradients with respect to the logits (i.e., $\mathcal{A}^t(\textcolor{blue}{x_v})$ and $\mathcal{G}^t(\textcolor{red}{x_u})$) are typically bounded, this implies that the relative interference is the dominant factor driving forgetting. A larger $||\mathcal{K}^t||_F$ means more interference between $\textcolor{red}{x_u}$ and $\textcolor{blue}{x_v}$. Besides, our analysis in this section also depends on the assumption of the ''eNTK matrix $\mathcal{K}^t$ remains roughly stable over training'', which is well-validated in [1] and our following experiments.*"
>
> [1] *Yi Ren and Danica J Sutherland. Learning dynamics of llm finetuning. ICLR 2025.*
>
> **Weaknesses 4:**
>
> Thanks, we have revised the text and introduced $A_{i,t}$ as follows:
>
> "*$\mathbf{r}=\{r_1,\cdots,r_G\}$ is reward for model outputs $\{o_1,\cdots,o_G\}$, $A_{i,t}=(r_i-\text{mean}(\mathbf{r}))/\text{std}(\mathbf{r})$ is the advantage for each token.*"
>
> ### **Questions** ###
> **Question 1:**
>
> Thanks. In fact, the difference in training steps does not create an unfair comparison. Our focus is on the amount of forgetting when the two methods achieve similar performance on the new task, not on their learning efficiency.
>
> For both RFT and SFT, increasing the number of training steps typically leads to more forgetting of old knowledge. In our experiments, even though RFT uses many more training steps than SFT, it still achieves comparable performance on the new task while exhibiting far less forgetting. This implies that giving SFT additional steps would only make its forgetting worse.
>
> Moreover, **Fig. 5** presents the Pareto front of the two methods, showing that under matched conditions, RFT consistently demonstrates better forgetting–performance trade-offs than SFT.
>
> **Question 2:**
>
> Thanks, the stability of eNTK during fine-tuning has already been extensively validated in prior work (**Appendix C** of [1]). In our jigsaw experiments, we also confirm this behavior empirically. As shown in **Fig. 2**, during SFT fine-tuning, the lower bound of the eNTK norm quickly stabilizes after only a few dozen training steps.
>
> We also include additional fine-tuning results on more models and tasks in the newly added **Sec 5.7**, and they show the same pattern: the eNTK (**Fig. 6**) becomes stable very quickly.
>
> [1] *Yi Ren and Danica J Sutherland. Learning dynamics of llm finetuning. ICLR 2025.*
>
> **Question 3:**
>
> They are consistent. We ensured that the only difference between Rea-GRPO-Rollout and Rea-4o-Rollout is how the intermediate CoT is generated. All other settings, including the SFT prompts and the way answers are computed, are kept exactly the same.

---

> ### Author Response · Authors · 2025-11-27
> **Invitation to discussion**
>
> Dear Reviewer sRUL,
>
> Thank you again for your thoughtful review. To address your concerns, we:
> - Added **new experiments on Qwen2.5-3B/7B-Instruct (math: ORZ, GSM8K, MATH-500, IFEval)** to demonstrate that our findings go beyond multimodal jigsaw.
> - Clarified that our goal is to **analyze why RFT forgets less than SFT**, and expanded the **CL-related work** (EWC, LwF, ER, etc.).
> - **Rewrote Sec. 5.4** to give a more intuitive explanation of the learning dynamics and eNTK (including the definition of \(A_{i,t}\) and the eNTK stability assumption), and discussed **training-step fairness** via a Pareto front view.
> - Updated the PDF, where **all new or revised content is highlighted in purple** for easy checking.
>
> We hope these responses have addressed your concerns and would appreciate it if you could consider raising your final rating. We look forward to your further feedback!
>
> Best regards,
> The Authors

---

### Official Review · Reviewer_Ssap · 2025-10-28

**Soundness:** 2
**Presentation:** 2
**Contribution:** 2
**Rating:** 6
**Confidence:** 4

**Summary:**

This paper investigates how Supervised Fine-Tuning (SFT) and Reinforcement Fine-Tuning (RFT) affect prior knowledge retention in Multimodal Large Language Models (MLLMs), using jigsaw puzzles as a novel task. It finds that SFT causes severe catastrophic forgetting despite rapid task learning, whereas RFT learns more slowly but preserves prior knowledge effectively. The key insight is that the data distribution, not the algorithm itself, drives forgetting; RFT naturally generates low-perplexity, model-aligned training samples that interfere less with existing knowledge.

**Strengths:**

1. The forgetting is less expored in multimodal setting.
2. Provide theorectical analysis.

**Weaknesses:**

1. Limited Novelty of Core Insight
Given that [1] has already demonstrated that reinforcement learning can effectively mitigate catastrophic forgetting, the contribution of this work appears limited. In fact, [1] also conducted experiments showing that it is the on-policy training paradigm (e.g., RFT) that helps overcome catastrophic forgetting. The key distinction between the data used in SFT and RFT lies in whether they are on-policy or offline. Beyond the discussion in L153–L157, the authors should provide a more detailed comparison between their study and [1]. To me, this paper merely provides a possible explanation for why using on-policy data can help overcome catastrophic forgetting, rather than being the one that discovers this phenomenon.
- In L55, the paper claims that RFT can “discover novel and useful problem-solving strategies.” However, I find it difficult to draw this conclusion from the preceding discussion about RFT, which merely states that “RFT requires several tens of thousands of training steps to successfully solve jigsaw puzzles.”
- The paper lacks a detailed description of the construction process of the jigsaw puzzle task, such as the method used to shuffle the image tiles, and it also does not report the detailed training cost of the experiments. This omission makes it difficult for other researchers to reproduce the results.

[1] Rl’s razor: Why online reinforcement learning forgets less

2. Unjustified Focus on Multimodality
While the study is conducted using multimodal models (MLLMs), the analysis and conclusions are not inherently tied to multimodal reasoning. The mechanisms explored—such as data distribution and perplexity—are equally applicable to unimodal language models, raising questions about the necessity of the multimodal framing.

3. Opaque Theoretical Framework
The theoretical analysis, based on learning dynamics and neural tangent kernels, is presented with limited intuitive explanation or clear connection to the empirical findings. This obscures how the theory substantiates the main claims and reduces its accessibility to a broader audience.

**Questions:**

- In eq 1, why the author set $\pi_{\theta_{\mathrm{old}}}(\cdot)=\pi_\theta(\cdot)$, instead of using the standard GRPO recipe? Can you provide the experiment result using the standard GRPO recipe?
- In the rationale shown in Figure 7, I noticed that the model’s reasoning process differs from how humans typically solve a jigsaw puzzle. As a human, I do begin by identifying the overall scene depicted in the image—similar to the reasoning shown in Figure 7—by observing the misaligned image tiles. However, my step-by-step reasoning primarily involves progressively determining the position of each tile. Once a tile’s position is confirmed, humans usually infer the positions of adjacent tiles based on the edge pixels of the confirmed piece. In contrast, the rationale presented in Figure 7 does not capture this gradual confirmation process. Please analyze the underlying reason for this discrepancy.

---

> ### Author Response · Authors · 2025-11-20
> **Comment of Reviews by Reviewer Ssap [1/2]**
>
> ### **Weaknesses** ###
>
> **Weaknesses 1 (Novelty):**
>
> Firstly, we would like to point out that [1] is a very recent work released on ArXiv in September. It is a contemporary work. According to official Reviewer Guide of ICLR 2026:"authors are not required to compare to contemporaneous work or unpublished arxiv papers".
>
> Secondly, [1] only observes that RL forgets less than SFT with smaller KL divergence and RL leads to smaller KL shifts with on-policy data. [1] donot tell why on-policy data interference less prior knowledge. However, our paper not only discovers this phenomenon of RFT, but also proposes a learning dynamics based interpretation of forgetting that decomposes how training data influence prior knowledge into its magnitude and direction. Based on this novel perspective of learning dynamics, we are able to curate better dataset for post-training that unifies the advantage of SFT's fast adaption and RFT's preservation of prior knowledge (which is originally discussed in our **Sec. 6**. We now supplied some initial experiments about this point on **Sec 5.8**). Further, our paper also first shows that large-scale RFT can solve unseen tasks that beyond the capacity boundary of base model.
>
> Thirdly, based on the actual timeline of our experiments, we discover the forgetting phenomenon of RFT several months before the ICLR submission deadline. After noticing this phenomenon, we further carried out extensive interpretability analysis and experiments, which eventually led us to propose a learning dynamics based framework. Therefore, it is not accurate to say that *“this paper merely provides ..., rather than being the one that discovers the phenomenon.”* The fact that [1] appeared on arXiv does not reflect the actual discovery timeline of our work.
>
> [1] *Rl’s razor: Why online reinforcement learning forgets less*
>
>
> **Weaknesses 1 (L55):**
>
> Thanks, what we mean is that large-scale RL training enables the base model to improve from essentially 0% accuracy to 60% on unseen tasks, which is a capability of RFT that, to the best of our knowledge, has not been shown in prior work. Earlier studies typically start with models that already possess mathematical or coding abilities, and then focus on further improving their reasoning skills. In our sentence, “novel and useful problem-solving strategies” specifically refers to this ability to learn from 0% to 60% accuracy. We have already revised this sentence to make its meaning clearer and avoid ambiguity in **L55**.
>
> **Weaknesses 1 (Experiment Details):**
>
> We have provided the construction process of the jigsaw puzzle task in **Sec. 3.1**. We have also provided the GPU type and count, number of training steps, and global batch size in the **Section. 4** (Hyper-parameter setup). Here, we provide a more detailed description of our experiments.
>
> Our $3\times3$ jigsaw dataset is built upon COCO images as follows. For each image, we resize it by bicubic interpolation to the nearest resolution divisible by 3. Then we slice the resized image into a $3\times3$ grid and assign row-major indices $k \in \{0,\dots,8\}$ to tiles. We use a uniform random permutation of ${0,\dots,8}$ to get the shuffled tiles. At training time, the model receives the shuffled tiles and outputs the canonical top-left–to–bottom-right indices. Besides, we have added a **pseudocode** of the construction process of jigsaw puzzles in **Algorithm 1** of the Appendix in the revised paper.
>
> Regarding the training cost, we provide the total GPU hours (Number of GPU $\times$ Training Hours) for the main experiment configurations of Jigsaw Puzzles and Math Reasoning below. The largest configuration (Qwen2.5-VL-7B (jigsaw) RFT) requires about 2200 GPU-hours, while the SFT is two orders of magnitude cheaper.
>
> Finally, we will also release the preprocessing script, training codes, and Jigsaw dataset to support reproduction.
>
> **Table: Total training cost (in GPU-hours) for different model sizes and training recipes for the jigsaw puzzles (Qwen2.5-VL-3B/7B) and math reasoning (Qwen2.5-3B/7B).**
> | Method                               | Qwen2.5-VL-3B (jigsaw) | Qwen2.5-VL-7B (jigsaw) | Qwen2.5-3B (math) | Qwen2.5-7B (math) |
> |--------------------------------------|------------------------:|------------------------:|------------------:|------------------:|
> | RFT                                 |                   710   |                  2200   |              72   |              96   |
> | SFT-Non-Rea                          |                   2.3   |                     4   |            0.77   |            1.33   |
> | SFT-Rea-4o-Rollout                   |                   6.4   |                  11.5   |             1.3   |            2.27   |
> | SFT-Rea-GRPO-Rollout                 |               5.1 |             8.3  |        1.27  |        2.47  |
>
> **Weaknesses 2:**
>
> Thanks, Please refer to our **General Comment** for More Experiments on Other Tasks and Models.

---

> ### Author Response · Authors · 2025-11-20
> **Comment of Reviews by Reviewer Ssap [2/2]**
>
> **Weaknesses 3:**
>
> Thanks, our theoretical analysis is supported by consistent empirical evidence. Learning dynamics describe how changes in specific factors affect model predictions. Motivated by this idea, we use learning dynamics to characterize how learning new knowledge influences previously learned knowledge. The eNTK between new and old knowledge naturally measures the strength of their interference, while the PPL of the new knowledge under the base model reflects how compatible the new knowledge is with the old.
>
> In **Fig. 2**, we observe that the Non-Rea data have a very strong interference with old knowledge, which aligns with the severe catastrophic forgetting shown in **Tab. 1**. In contrast, the reasoning data (Rea-GRPO-Rollout and Rea-4o-Rollout) show much smaller eNTK values. Beyond that, **Fig. 4** further shows a clear correspondence between their PPL values and the degree of forgetting, consistent with **Theorem 5.2**: the interference between samples is mutual and symmetric. Data with lower PPL benefit more from old knowledge during pretraining and, by **Theorem 5.2**, strengthening such data during RL also tends to benefit old knowledge. This is exactly what we observe in **Fig. 8**, where PPL of old knowledge decreases more clearly during Rea-GRPO-Rollout fine-tuning than during Rea-4o-Rollout.
>
> Moreover, in the **General Comment**, we provide additional experiments on other tasks and models. Their eNTK and PPL trends **match our earlier observations**, further confirming the correctness and usefulness of our theory.
>
> ### **Questions** ###
>
> **Question 1:**
>
> Thanks, in our initial experiments, we followed the default settings of the HuggingFace/trl framework, where num_iterations is set to 1 (i.e., the GRPO parameter $\mu$). This means that, by default, the two models ($\pi_{\theta_{\text{old}}}(\cdot)$ and $\pi_{\theta}(\cdot)$) are identical.
>
> In the table below, we report RFT results on math reasoning with different GRPO training recipes. The experiments show that using the standard GRPO recipe and our recipe yield only minimal differences in model performance on new/old tasks.
>
> **Table: Performance comparison of different GRPO training recipe.**
> | Task                 | 3B-Base  | 3B-RFT (μ=1) | 3B-RFT (μ=2) | 3B-RFT (μ=4) | 7B-Base  | 7B-RFT (μ=1) | 7B-RFT (μ=2) | 7B-RFT (μ=4) |
> |----------------------|----------|--------------|--------------|--------------|----------|--------------|--------------|--------------|
> | ORZ Test (New)       | **21.32** | 35.00        | 35.70        | 34.10        | **32.11** | 49.29        | 48.18        | 47.12        |
> | GSM8k (Old)          | **84.08** | 83.40        | 83.02        | 81.27        | **90.14** | 90.22        | 90.29        | 89.69        |
> | Math-500 (Old)       | **42.40** | 55.20        | 55.40        | 52.80        | **66.60** | 64.80        | 65.20        | 64.00        |
> | IFEval (Old)         | **71.58** | 73.38        | 74.82        | 73.98        | **80.58** | 80.46        | 81.89        | 80.46        |
>
> **Question 2:**
>
> Thanks. Because RFT does not provide any supervision on the Chain-of-Thought (CoT), and the base model itself has no inherent ability to solve jigsaw tasks, RFT can only guarantee an improvement in the final *accuracy* of the jigsaw. The CoT produced during RFT is simply a pattern the model discovers through repeated rollouts that helps increase accuracy; it is not guaranteed to be correct or aligned with human reasoning.
>
> If the goal is to obtain human-like reasoning processes, an additional continual pre-training or mid-training stage would likely be needed. This stage would explicitly inject jigsaw-related data, such as edge comparison based corpora, so that the model is able to generate more coherent and human-like CoT during RFT.

---

> ### Author Response · Authors · 2025-11-27
> **Invitation to discussion**
>
> Dear Reviewer Ssap,
>
> Thank you again for your thoughtful review. To address your concerns, we:
> - Added **new experiments on Qwen2.5-3B/7B-Instruct (math: ORZ, GSM8K, MATH-500, IFEval)** to demonstrate that our findings go beyond multimodal jigsaw.
> - Clarified our **novelty beyond “RL’s razor”** (learning-dynamics view + data-design perspective) and the discovery timeline.
> - Provided more details on **jigsaw construction and training cost** (pseudocode, GPU-hours), and reported **different GRPO recipes**, as well as a discussion of why RFT CoT may differ from human reasoning.
> - Updated the PDF, where **all new or revised content is highlighted in purple** for easy checking.
>
> We hope these responses have addressed your concerns and would appreciate it if you could consider raising your final rating. We look forward to your further feedback!
>
> Best regards,
> The Authors

---

### Author Response · Authors · 2025-11-20
**General Comment for All Reviewers [1/3]**

### **Paper Revision:** ###

Thanks to all reviewers' insightful reviews, we have revised the paper accordingly. Concerns are addressed below and in the revised paper, all the revised parts in the paper have been highlighted in purple.

### **General Comment for More Experiments on Other Tasks and Models (Math Reasoning):** ###

Thank you for the reviewers’ suggestions. Our theory is indeed **not limited** to MLLMs. The mechanisms we study, such as data distribution and perplexity, apply equally well to unimodal language models and other tasks.

To support this point, we additionally provide experiments on the LLM Qwen2.5-Instruct here and in the revised paper, showing its forgetting behavior during post-training on math reasoning, along with the corresponding results. We hope these extra experiments can further **strengthen the generality and credibility** of our theoretical analysis and conclusions.

Note that all figures mentioned here can be found in the revised paper.

***Math Reasoning Dataset.***

We use the curated math corpus released by Open-Reasoner-Zero[1] as our large-scale reasoning-oriented training data, and randomly split it into $90\%$ training and $10\%$ held-out test data.

We refer to this held-out split as *ORZ Test*, which serves as a new target task for post-training.

Each example is a competition-style math problem paired with a verifiable final answer, without any visual input.

[1] *Hu J, Zhang Y, Han Q, et al. Open-reasoner-zero: An open source approach to scaling up reinforcement learning on the base model[J]. arXiv preprint arXiv:2503.24290, 2025.*

***LLMs and Evaluation.***

For math experiments, we use Qwen2.5-3B-Instruct and Qwen2.5-7B-Instruct as base LLMs. We treat ORZ Test as a new target task and report answer accuracy on it, and use GSM8K, MATH-500, and IFEval as prior knowledge to monitor retention of prior math and instruction-following abilities, with their standard accuracy metrics.

***Results.***

As summarized in below Table, the math reasoning experiments exhibit a forgetting pattern highly consistent with our multimodal jigsaw setting: on both 3B and 7B scales, SFT-Non-Rea achieves the largest performance drop on the old math (GSM8K, MATH-500) and instruction-following (IFEval) benchmarks, while reasoning-augmented SFT with external CoT (SFT-Rea-4o-Rollout) forgets less but still substantially more than SFT-Rea-GRPO-Rollout. The latter attains strong gains on the new ORZ task while keeping the performance on old tasks close to the base models, indicating the same hierarchy of forgetting severity, i.e., Non-Rea $>$ Rea-4o $>$ Rea-GRPO.

**Table: Performance comparison across Qwen2.5 models (math reasoning & instruction following).**
| Task / Info        | 3B-Base  | 3B-RFT | 3B-SFT-Non-Rea | 3B-SFT-Rea-4o-Rollout | 3B-SFT-Rea-GRPO-Rollout | 7B-Base  | 7B-RFT | 7B-SFT-Non-Rea | 7B-SFT-Rea-4o-Rollout | 7B-SFT-Rea-GRPO-Rollout |
|--------------------|----------|--------|-----------------|------------------------|-------------------------|----------|--------|-----------------|------------------------|-------------------------|
| Training steps     | --       | 2,650  | 1,600           | 2,140                  | 2,140                   | --       | 2,650  | 1,600           | 2,140                  | 2,140                   |
| ORZ Test (New)     | **21.3** | **35.0**   | 23.4            | 35.4                   | **37.8**                    | **32.1** | **49.3**   | 30.3            | 45.0                   | **53.4**                    |
| GSM8k (Old)        | **84.1** | **83.4**   | 15.1            | 79.9                   | **83.0**                    | **90.1** | **90.2**   | 21.8            | 85.8                   | **90.3**                    |
| Math-500 (Old)     | **42.4** | **55.2**   | 19.4            | 50.8                   | **54.4**                    | **66.6** | **64.8**   | 26.4            | 57.2                   | **66.4**                    |
| IFEval (Old)       | **71.6** | **73.4**   | 64.0            | 68.0                   | **72.7**                    | **80.6** | **80.5**   | 57.2            | 64.4                   | **80.0**                    |
***Analysis.***

To probe the underlying mechanism, we compute LBK between post-training samples and prior math knowledge during SFT. As shown in **Fig. 6** of revised paper, Non-Rea data consistently display much larger LBK values and Rea-4o also contains occasional high-LBK outliers, providing further evidence for the generality of our learning-dynamics analysis in **Sec. 5.4**. Moreover, **Fig. 7** of revised paper shows that Rea-4o-Rollout is concentrated in the high-perplexity region of the base models, whereas Rea-GRPO-Rollout lies closer to the low-perplexity region, mirroring our findings on jigsaw puzzles and supporting the low-perplexity training hypothesis in **Sec. 5.5** that post-training on model-aligned (low-PPL) reasoning trajectories mitigates catastrophic forgetting.

---

> ### Author Response · Authors · 2025-11-20
> **General Comment for All Reviewers [2/3]**
>
> ***Pareto Frontier of SFT-Rea-GRPO-Rollout and SFT-Rea-4o-Rollout.***
>
> We further sweep the learning rate over $\{1\times10^{-6}, 5\times10^{-6}, 1\times10^{-5}, 2\times10^{-5}\}$ and, for each setting, plot the Pareto-optimal frontier between performance on the new ORZ task and the performance on old tasks (GSM8K, MATH-500, and IFEval) in **Fig. 13** of revised paper. Across all learning rates and both 3B/7B scales, SFT training on Rea-GRPO-Rollout consistently achieves a strictly better Pareto frontier than on Rea-4o-Rollout, yielding either higher ORZ accuracy under a similar level of forgetting, or better retention of prior knowledge at comparable ORZ performance.
>
> ***Perplexity on Prior knowledge during SFT.***
>
> **Fig. 14** of the revised paper also shows that training on Rea-GRPO-Rollout maintains the perplexity of old math corpora much more stably than Rea-4o-Rollout, these results further corroborate our low-perplexity training hypothesis in **Sec. 5.5** and demonstrate that the advantages of Rea-GRPO-Rollout are robust under different optimization hyperparameters.

---

> ### Author Response · Authors · 2025-12-01
> **General Comment for All Reviewers [3/3]**
>
> ### **General Comment for More Experiments on Other Tasks and Models (Scientific QA):** ###
>
> To further demonstrate that our conclusions are **not limited** to multimodal jigsaw or math-only settings, we conduct additional experiments on a scientific multiple-choice QA benchmark, **Sci-MCQ4** from *SciKnowEval*[1], using **Qwen2.5-3B-Instruct** as the base LLM. The new results and figures are included in the **revised paper** and highlighted in purple.
>
> ***Setup.***
>
> We randomly sample *8,500* Sci-MCQ4 examples and split them into 90% training / 10% held-out test; the held-out split (*Sci-MCQ4 Test*) is treated as the **new target task**. We reuse **GSM8K, MATH-500, and IFEval** as **old tasks** to monitor retention of prior math and instruction-following abilities.
>
> ***Results.***
>
> | Model                    | Steps | Sci-MCQ4 (New) | GSM8K (Old)    | MATH-500 (Old) | IFEval (Old)   |
> |--------------------------|-------|----------------|----------------|----------------|----------------|
> | Base                     |  --   | 65.1           | 84.1           | 42.4           | 71.6           |
> | RFT                      |  960  | 70.8 (+5.7)    | 82.9 (−1.1)    | 45.2 (+2.8)    | 71.8 (+0.2)    |
> | SFT-Non-Rea              |  240  | 69.3 (+4.2)    | 69.4 (**−14.6**)   | 43.2 (+0.8)    | 70.3 (**−1.3**)    |
> | SFT-Rea-4o-Rollout       |  214  | 67.0 (+1.9)    | 74.1 (**−9.9**)    | 40.2 (−2.2)    | 73.9 (+2.3)    |
> | SFT-Rea-GRPO-Rollout     |  214  | 71.7 (**+6.6**)    | 81.3 (−2.8)    | 40.8 (−1.6)    | 74.5 (+2.9)    |
> ---
> ***Analysis.***
>
> All post-training methods improve Sci-MCQ4 accuracy over the base model, with **SFT-Rea-GRPO-Rollout** achieving the largest gain (+6.6 points) while largely preserving performance on GSM8K, MATH-500, and IFEval. In contrast, **SFT-Non-Rea** exhibits the strongest degradation on GSM8K and IFEval, and **SFT-Rea-4o-Rollout** lies in between, again yielding a consistent hierarchy of forgetting severity: **Non-Rea > Rea-4o > Rea-GRPO** on a scientific QA domain. The corresponding LBK and perplexity analyses (**Fig. 16** and **Fig.17** in the revised paper) show the same pattern as in our math and jigsaw experiments, further supporting the low-perplexity training hypothesis.
>
> [1] *Kehua Feng, Keyan Ding, Weijie Wang, Xiang Zhuang, Zeyuan Wang, Ming Qin, Yu Zhao, Jianhua Yao, Qiang Zhang, and Huajun Chen. Sciknoweval: Evaluating multi-level scientific knowledge of large language models. arXiv preprint arXiv:2406.09098, 2024.*

---

### Meta-Review · Area_Chair_M21k · 2025-12-26

**Summary:**

Summary of major concerns (in order of importance):

1. Reviewers sRUL, DcMQ: Limited empirical evidence; the original experiments were only on the Qwen 2.5 VL model at 3B and 7B sizes, and only on one task (jigsaw puzzles).
2. Reviewers Ssap, sRUL: Theory is too dense and an intuitive connection to the experiments is missing.
3. Reviewer Ssap: Limited novelty, especially compared to Shenfeld et al. (2025).
4. Reviewer DcMQ: Concerns whether a simpler data diversity based intervention would have better bridged the gap between RFT and SFT.

**Reviewer Concerns:**

Addressed concerns:

Of the major concerns listed in the previous section, I think 3 and 4 are fully addressed by the rebuttal. For 3, the authors pointed out that the relevant paper was posted on ArXiv only a few weeks before the ICLR deadline, and for 4, they presented new results which show that increasing the data diversity alone is insufficient to bridge the gap. In addition, some additional concerns around efficiency (DcMQ), missing data details (Ssap), and missing related work (sRUL) which were also addressed well by the rebuttal.

For concern 2, the authors included additional explanation in the paper, which addresses the concern to some extent.

Outstanding concerns:

For concern 1, the authors included new experiments with Qwen 2.5 Instruct 3B and 7B, and on more tasks, ORZ, GSM-8K, MATH, IFEval. While these experiments do make the paper stronger, it would be more convincing to see experiments with models from a different family.

**Reviewer Scores:**

The new results presented in the discussion period directly address some of the concerns from sRUL and DcMQ. So I think they would have increased their scores to 6 or 8. I find the response to Ssap's concern regarding novelty convincing too, so there is a chance that score would have been increased too.

---

### Decision · Program_Chairs · 2026-01-26

Accept (Poster)